# Block Sparse Bayesian Learning: A Diversified Scheme

**Yanhao Zhang**    **Zhihan Zhu**    **Yong Xia** [*]
School of Mathematical Sciences, Beihang University
Beijing, 100191
{yanhaozhang, zhihanzhu, yxia}@buaa.edu.cn

## Abstract

This paper introduces a novel prior called Diversified Block Sparse Prior to characterize the widespread block sparsity phenomenon in real-world data. By allowing diversification on intra-block variance and inter-block correlation matrices, we effectively address the sensitivity issue of existing block sparse learning methods to pre-defined block information, which enables adaptive block estimation while mitigating the risk of overfitting. Based on this, a diversified block sparse Bayesian learning method (DivSBL) is proposed, utilizing EM algorithm and dual ascent method for hyperparameter estimation. Moreover, we establish the global and local optimality theory of our model. Experiments validate the advantages of DivSBL over existing algorithms.

## 1   Introduction

Sparse recovery through Compressed Sensing (CS), with its powerful theoretical foundation and broad practical applications, has received much attention [1]. The basic model is considered as

$$\mathbf{y} = \mathbf{\Phi}\mathbf{x}, \tag{1}$$

where $\mathbf{y} \in \mathbb{R}^{M \times 1}$ is the measurement (or response) vector and $\mathbf{\Phi} \in \mathbb{R}^{M \times N}$ is a known design matrix, satisfying the Unique Representation Property (URP) condition [2]. $\mathbf{x} \in \mathbb{R}^{N \times 1}(N \gg M)$ is the sparse vector to be recovered. In practice, $\mathbf{x}$ often exhibits transform sparsity, becoming sparse in a transform domain such as Wavelet, Fourier, etc. Once the signal is compressible in a linear basis $\Psi$, in other words, $\mathbf{x} = \mathbf{\Psi}\mathbf{w}$ where $\mathbf{w}$ exhibits sparsity, and $\mathbf{\Phi}\mathbf{\Psi}$ satisfies Restricted Isometry Constants (RIP) [3], then we can simply replace $\mathbf{x}$ by $\mathbf{\Psi}\mathbf{w}$ in (1) and solve it in the same way. Classic algorithms for compressive sensing and sparse regression include Lasso [4], Sparse Bayesian Learning (SBL) [5], Basis Pursuit (BP) [6], Orthogonal Matching Pursuit(OMP) [7], etc. Recently, there have been approaches that involve solving CS problems through deep learning [8; 9; 10].

However, deeper research into sparse learning has shown that relying solely on the sparsity of $\mathbf{x}$ is insufficient, especially with limited samples [11; 12]. Widely encountered real-world data, such as image and audio, often exhibit clustered sparsity in transformed domains [13]. This phenomenon, known as block sparsity, means the sparse non-zero entries of $\mathbf{x}$ appear in blocks [11]. Recent years, block sparse models have gained attention in machine learning, including sparse training [14], adversarial learning [15], image restoration [16; 14], (audio) signal processing [17; 18] and many other areas. Generally, the block structure of $\mathbf{x}$ with $g$ blocks is defined by

$$\mathbf{x} = [\underbrace{x_1 \ldots x_{d_1}}_{\mathbf{x}_1^T} \underbrace{x_{d_1+1} \ldots x_{d_1+d_2}}_{\mathbf{x}_2^T} \cdots \underbrace{x_{N-d_g+1} \ldots x_N}_{\mathbf{x}_g^T}]^T, \tag{2}$$

where $d_i(i = 1...g)$ represent the size of each block, which are not necessarily identical. Suppose only $k(k \ll g)$ blocks are non-zero, indicating that $\mathbf{x}$ is block sparse. Up to now, several methods have been proposed to recover block sparse signals. They are mainly divided into two categories.

[*]Corresponding author

38th Conference on Neural Information Processing Systems (NeurIPS 2024).

**Block-based** Classical algorithms for processing block sparse scenarios include Group-Lasso [19; 20; 21], Group Basis Pursuit [22], Block-OMP [11]. Blocks are assumed to be static with a fixed preset size. Furthermore, Temporally-SBL (TSBL) [23] and Block-SBL (BSBL) [24; 25], based on Bayesian models, provide refined estimation of correlation matrices within blocks. However, they assume elements within one block tend to be either zero or non-zero simultaneously. Although they can estimate intra-block correlation with high accuracy in block-level recovery, they require preset choices of suitable block sizes and patterns, which are too rigid for many practical applications.

**Pattern-based** StructOMP [26] is a pattern-based greedy algorithm allowing structures, which is a generalization of group sparsity. Another classic model named Pattern-Coupled SBL (PC-SBL) [27; 28], does not have a predefined requirement for block size as well. It utilizes a Bayesian model to couple the signal variances. Building upon PC-SBL, Burst PC-SBL, proposed in [29], is employed for the estimation of mMIMO channels. While pattern-based algorithms address the issue of explicitly specifying block patterns in block-based algorithms, these models provide a coarse characterization of the intra-block correlation, leading to a loss of structural information within the blocks.

In this paper, we introduce a diversified block sparse Bayesian framework that incorporates diversity in both variance within the same block and intra-block correlation among different blocks. Our model not only inherits the advantages of block-based methods on block-level estimation, but also addresses the longstanding issues associated with such algorithms: the diversified scheme reduces sensitivity to a predefined block size or specified block location, hence accommodates general block sparse data. Based on this model, we develop the DivSBL algorithm, and also analyze both the global minimum and local minima of the constrained cost function (likelihood). The subsequent experiments illustrate the superiority of proposed diversified scheme when applied to real-life block sparse data.

## 2 Diversified block sparse Bayesian model

We consider the block sparse signal recovery, or compressive sensing question in the noisy case

$$\mathbf{y} = \mathbf{\Phi}\mathbf{x} + \mathbf{n}, \tag{3}$$

where $\mathbf{n} \sim \mathcal{N}(0, \beta^{-1}\mathbf{I})$ represents the measurement noise, and $\beta$ is the precise scalar. Other symbols have the same interpretations as (1). The signal $\mathbf{x}$ exhibits block-sparse structure in (2), yet the block partition is unknown. For clarity in description, we assume that all blocks have equal size $L$, with the total dimension denoted as $N = gL$. Henceforth, we presume that the signal $\mathbf{x}$ follows the structure:

$$\mathbf{x} = [\underbrace{x_{11} \ldots x_{1L}}_{\mathbf{x}_1^T} \underbrace{x_{21} \ldots x_{2L}}_{\mathbf{x}_2^T} \cdots \underbrace{x_{g1} \ldots x_{gL}}_{\mathbf{x}_g^T}]^T. \tag{4}$$

In Sections 2.1.1 and 5.2, we clarify that this assumption is made without loss of generality. In fact, our algorithm can automatically adjust $L$ to an appropriate size, expanding or contracting as needed.

### 2.1 Diversified block sparse prior

The Diversified Block Sparse prior is proposed in the following scheme. Each block $\mathbf{x}_i \in \mathbb{R}^{L \times 1}$ is assumed to follow a multivariate Gaussian prior

$$p(\mathbf{x}_i; \{\mathbf{G}_i, \mathbf{B}_i\}) = \mathcal{N}(\mathbf{0}, \mathbf{G}_i\mathbf{B}_i\mathbf{G}_i), \forall i = 1, \cdots, g, \tag{5}$$

in which, $\mathbf{G}_i$ represents the Diversified Variance matrix, and $\mathbf{B}_i$ represents the Diversified Correlation matrix, with detailed formulations in Sections 2.1.1 and 2.1.2. Therefore, the prior distribution of the entire signal $\mathbf{x}$ is denoted as

$$p(\mathbf{x}; \{\mathbf{G}_i, \mathbf{B}_i\}_{i=1}^g) = \mathcal{N}(\mathbf{0}, \mathbf{\Sigma}_0), \tag{6}$$

where $\mathbf{\Sigma}_0 = \text{diag}\{\mathbf{G}_1\mathbf{B}_1\mathbf{G}_1, \mathbf{G}_2\mathbf{B}_2\mathbf{G}_2, \cdots, \mathbf{G}_g\mathbf{B}_g\mathbf{G}_g\}$. The dependency in this hierarchical model is shown in Figure.1.

#### 2.1.1 Diversified intra-block variance

We first execute diversification on variance. In (5), $\mathbf{G}_i$ is defined as

$$\mathbf{G}_i \triangleq \text{diag}\{\sqrt{\gamma_{i1}}, \cdots, \sqrt{\gamma_{iL}}\}, \tag{7}$$

and $\mathbf{B}_i \in \mathbb{R}^{L \times L}$ is a positive definite matrix capturing the correlation within the $i$-th block. According to the definition of Pearson correlation, the covariance term $\mathbf{G}_i\mathbf{B}_i\mathbf{G}_i$ in (5) can be specified as

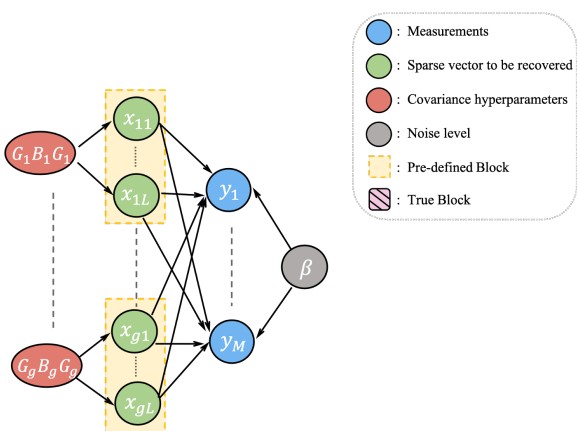

Figure 1: Directed acyclic graph of diversified block sparse hierarchical structure. Except for Measurements (blue nodes), which are known, all other nodes are parameters to estimate.

Figure 2: The gold dashed line shows the preset block, and the black shadow represents the actual position of the block with its true size.

$$\mathbf{G}_i\mathbf{B}_i\mathbf{G}_i = \begin{bmatrix} \gamma_{i1} & \rho_{12}^i\sqrt{\gamma_{i1}}\sqrt{\gamma_{i2}} & \cdots & \rho_{1L}^i\sqrt{\gamma_{i1}}\sqrt{\gamma_{iL}} \\ \rho_{21}^i\sqrt{\gamma_{i2}}\sqrt{\gamma_{i1}} & \gamma_{i2} & \cdots & \rho_{2L}^i\sqrt{\gamma_{i2}}\sqrt{\gamma_{iL}} \\ \vdots & \vdots & \ddots & \vdots \\ \rho_{L1}^i\sqrt{\gamma_{iL}}\sqrt{\gamma_{i1}} & \rho_{L2}^i\sqrt{\gamma_{iL}}\sqrt{\gamma_{i2}} & \cdots & \gamma_{iL} \end{bmatrix},$$

where $\rho_{sk}^i (\forall s, k = 1\cdots L)$ are the elements in correlation matrix $\mathbf{B}_i$, serving as a visualization of covariance with displayed structural information.

Now it is evident why assuming equal block sizes $L$ is insensitive. For the sake of clarity, we denote the true size of the $i$-th block $\mathbf{x}_i$ as $L_T^i$. As illustrated in Figure 2, when the true block falls within the preset block, the variances $\gamma_{i.}$ corresponding to the non-zero positions in $\mathbf{x}_i$ will be learned as non-zero values through posterior inference, while the variances at zero positions will automatically be learned as zero. When the true block is positioned across the preset block, several blocks of the predefined size $L$ covered by the actual block will be updated together, and likewise, variances will be learned as either zero or non-zero. In this way, both of the size and location of the blocks will be automatically learned through posterior inference on the variances.

### 2.1.2 Diversified inter-block correlation

Due to limited data and excessive parameters in intra-block correlation matrices $\mathbf{B}_i(\forall i)$, previous works correct their estimation by imposing strong correlated constraints $\mathbf{B}_i = \mathbf{B}(\forall i)$ to overcome overfitting [24]. Recognizing that correlation matrices among different blocks should be diverse yet still exhibit some correlation, we apply a weak-correlated constraint to diversify $\mathbf{B}_i$ in the model.

Here we introduce novel weak constraints on $\mathbf{B}_i$, specifically,

$$\psi(\mathbf{B}_i) = \psi(\mathbf{B}) \quad \forall i = 1, \cdots, g, \tag{8}$$

where $\psi : \mathbb{R}^{L^2} \to \mathbb{R}$ is the weak constraint function and $\mathbf{B}$ is obtained from the strong constraints $\mathbf{B}_i = \mathbf{B}(\forall i)$, as detailed in Section 3.2. Weak constraints (8) not only capture the distinct correlation structure but also avoid overfitting issue arising from the complete independence among different $\mathbf{B}_i$.

Furthermore, the constraints imposed here not only maintain the global minimum property of our algorithm, as substantiated in Section 4, but also effectively enhance the convergence rate of the algorithm. There are actually $gL(L+1)/2$ constraints in the strong correlated constraints $\mathbf{B}_i = \mathbf{B}(\forall i)$, while with (8), the number of constraints significantly decreases to $g$, yielding acceleration on the convergence rate. The experimental result is shown in Appendix A.

In summary, the prior based on (5), (6), (7) and (8) is defined as **diversified block sparse prior**.

### 2.1.3 Connections to classical models

Note that the classical Sparse Bayesian Learning models, Relevance Vector Machine (RVM) [5] and Block Sparse Bayesian Learning (BSBL) [23], are special cases of our model.

**Connection to RVM** Taking $\mathbf{B}_i$ as identity matrix, diversified block sparse prior (6) immediately degenerates to RVM model

$$p\left(x_i; \gamma_i\right) = \mathcal{N}\left(0, \gamma_i\right), \forall i = 1, \cdots, N, \tag{9}$$

which means ignoring the correlation structure.

**Connection to BSBL** When $\mathbf{G}_i$ is scalar matrix $\sqrt{\gamma_i}\mathbf{I}$, the formulation (5) becomes

$$p\left(\mathbf{x}_i; \{\gamma_i, \mathbf{B}_i\}\right) = \mathcal{N}\left(\mathbf{0}, \gamma_i \mathbf{B}_i\right), \forall i = 1, \cdots, g, \tag{10}$$

which is exactly BSBL model. In this case, all elements within a block share common variance $\gamma_i$.

## 2.2 Posterior estimation

By observation model (3), the Gaussian likelihood is

$$p(\mathbf{y} \mid \mathbf{x}; \beta) = \mathcal{N}(\mathbf{\Phi}\mathbf{x}, \beta^{-1}\mathbf{I}). \tag{11}$$

With prior (6) and likelihood (11), the diversified block sparse posterior distribution of $\mathbf{x}$ can be derived based on Bayes' theorem as

$$p\left(\mathbf{x} \mid \mathbf{y}; \{\mathbf{G}_i, \mathbf{B}_i\}_{i=1}^g, \beta\right) = \mathcal{N}\left(\boldsymbol{\mu}, \boldsymbol{\Sigma}\right), \tag{12}$$

where

$$\boldsymbol{\mu} = \beta \boldsymbol{\Sigma} \mathbf{\Phi}^T \mathbf{y}, \tag{13}$$

$$\boldsymbol{\Sigma} = \left(\boldsymbol{\Sigma}_0^{-1} + \beta \mathbf{\Phi}^T \mathbf{\Phi}\right)^{-1}. \tag{14}$$

After estimating all hyperparameters in (12), i.e, $\hat{\Theta} = \left\{\{\hat{\mathbf{G}}_i\}_{i=1}^g, \{\hat{\mathbf{B}}_i\}_{i=1}^g, \hat{\beta}\right\}$, as described in Section 3, the Maximum A Posterior (MAP) estimation of $\mathbf{x}$ is formulated as

$$\hat{\mathbf{x}}^{MAP} = \hat{\boldsymbol{\mu}}. \tag{15}$$

# 3 Bayesian inference: DivSBL algorithm

## 3.1 EM formulation

To estimate $\Theta = \{\{\mathbf{G}_i\}_{i=1}^g, \{\mathbf{B}_i\}_{i=1}^g, \beta\}$, either Type-II Maximum Likelihood [30] or Expectation-Maximization (EM) formulation [31] can be employed. Following EM procedure, our goal is to maximize $p(\mathbf{y}; \Theta)$, or equivalently $\log p(\mathbf{y}; \Theta)$. Defining objective function as $\mathcal{L}(\Theta)$, the problem can be expressed as

$$\max_{\Theta} \quad \mathcal{L}(\Theta) = -\mathbf{y}^T \boldsymbol{\Sigma}_y^{-1} \mathbf{y} - \log \det \boldsymbol{\Sigma}_y, \tag{16}$$

where $\boldsymbol{\Sigma}_y = \beta^{-1}\boldsymbol{I} + \mathbf{\Phi}\boldsymbol{\Sigma}_0\mathbf{\Phi}^T$. Then, treating $\mathbf{x}$ as hidden variable in E-step, we have $Q$ function as

$$\begin{aligned}
Q(\Theta) &= E_{x|y;\Theta^{t-1}}[\log p(\mathbf{y}, \mathbf{x}; \Theta)] \\
&= E_{x|y;\Theta^{t-1}}[\log p(\mathbf{y} \mid \mathbf{x}; \beta)] + E_{x|y;\Theta^{t-1}}\left[\log p\left(\mathbf{x}; \{\mathbf{G}_i\}_{i=1}^g, \{\mathbf{B}_i\}_{i=1}^g\right)\right] \\
&\triangleq Q(\beta) + Q(\{\mathbf{G}_i\}_{i=1}^g, \{\mathbf{B}_i\}_{i=1}^g),
\end{aligned} \tag{17}$$

where $\Theta^{t-1}$ denotes the parameter estimated in the latest iteration. As indicated in (17), we have divided $Q$ function into two parts: $Q(\beta) \triangleq E_{x|y;\Theta^{t-1}}[\log p(\mathbf{y} \mid \mathbf{x}; \beta)]$ depends solely on $\beta$, and $Q(\{\mathbf{G}_i\}_{i=1}^g, \{\mathbf{B}_i\}_{i=1}^g) \triangleq E_{x|y;\Theta^{t-1}}\left[\log p\left(\mathbf{x}; \{\mathbf{G}_i\}_{i=1}^g, \{\mathbf{B}_i\}_{i=1}^g\right)\right]$ only on $\{\mathbf{G}_i\}_{i=1}^g$ and $\{\mathbf{B}_i\}_{i=1}^g$. Therefore, the parameters of these two $Q$ functions can be updated separately.

In M-step, we need to maximize the above $Q$ functions to obtain the estimation of $\Theta$. As shown in Appendix B, the updating formula for $\gamma_{ij}, \mathbf{B}_i$ can be obtained as follows[2]:

$$\gamma_{ij} = \frac{4\mathbf{A}_{ij}^2}{(\sqrt{\mathbf{T}_{ij}^2 + 4\mathbf{A}_{ij}} - \mathbf{T}_{ij})^2}, \tag{18}$$

---

[2]Using MATLAB notation, $\boldsymbol{\mu}^i \triangleq \boldsymbol{\mu}((i-1)L + 1 : iL), \boldsymbol{\Sigma}^i \triangleq \boldsymbol{\Sigma}((i-1)L + 1 : iL, (i-1)L + 1 : iL)$.

$$\mathbf{B}_i = \mathbf{G}_i^{-1}\left(\mathbf{\Sigma}^i + \boldsymbol{\mu}^i\left(\boldsymbol{\mu}^i\right)^T\right)\mathbf{G}_i^{-1}, \tag{19}$$

where $\mathbf{T}_{ij}$ and $\mathbf{A}_{ij}$ are expressed as $\mathbf{T}_{ij} = \left[(\mathbf{B}_i^{-1})_{j\cdot} \odot \mathrm{diag}(\mathbf{W}_{-j}^i)^{-1}\right] \cdot \left(\mathbf{\Sigma}^i + \boldsymbol{\mu}^i(\boldsymbol{\mu}^i)^T\right)_{\cdot j}, \mathbf{A}_{ij} = (\mathbf{B}_i^{-1})_{jj} \cdot \left(\mathbf{\Sigma}^i + \boldsymbol{\mu}^i\left(\boldsymbol{\mu}^i\right)^T\right)_{jj},$ and $\mathbf{W}_{-j}^i = \mathrm{diag}\{\sqrt{\gamma_{i1}}, \cdots, \sqrt{\gamma_{i,j-1}}, 0, \sqrt{\gamma_{i,j+1}}, \cdots, \sqrt{\gamma_{iL}}\}$. The update formula of $\beta$ is derived in the same way as [23]. The learning rule is given by

$$\beta = \frac{M}{\|\mathbf{y} - \mathbf{\Phi}\boldsymbol{\mu}\|_2^2 + \mathrm{tr}\left(\mathbf{\Sigma}\mathbf{\Phi}^T\mathbf{\Phi}\right)}. \tag{20}$$

## 3.2 Diversified correlation matrices by dual ascent

Now we propose the algorithm for solving the correlation matrix estimation problem satisfying (8). As mentioned in Section 2.1.2, previous studies have employed strong constraints $\mathbf{B}_i = \mathbf{B}(\forall i)$, i.e.,

$$\mathbf{B} = \mathbf{B}_i = \frac{1}{g}\sum_{i=1}^{g}\mathbf{G}_i^{-1}\left(\mathbf{\Sigma}^i + \boldsymbol{\mu}^i\left(\boldsymbol{\mu}^i\right)^T\right)\mathbf{G}_i^{-1}. \tag{21}$$

In diversified scheme, we apply weak-correlated constraints (8) to diversify $\mathbf{B}_i$. Therefore, the problem of maximizing the $Q$ function with respect to $\mathbf{B}_i$ becomes

$$\begin{aligned}
\max_{\mathbf{B}_i} \quad & Q(\{\mathbf{B}_i\}_{i=1}^g, \{\mathbf{G}_i\}_{i=1}^g) \\
\mathrm{s.\,t.} \quad & \psi\left(\mathbf{B}_i\right) = \psi\left(\mathbf{B}\right) \quad \forall i = 1, \cdots, g,
\end{aligned} \tag{22}$$

which is equivalent to (in the sense that both share the same optimal solution)

$$\begin{aligned}
\min_{\mathbf{B}_i} \quad & \frac{1}{2}\log\det\mathbf{\Sigma}_0 + \frac{1}{2}\mathrm{tr}\left[\mathbf{\Sigma}_0^{-1}(\mathbf{\Sigma} + \boldsymbol{\mu}\boldsymbol{\mu}^T)\right] \\
\mathrm{s.\,t.} \quad & \psi\left(\mathbf{B}_i\right) = \psi\left(\mathbf{B}\right) \quad \forall i = 1, \cdots, g,
\end{aligned} \tag{P}$$

where $\mathbf{B}$ is already derived in (21). Therefore, by solving (P), we will obtain diversified solution for correlation matrices $\mathbf{B}_i, \forall i$. The constraint function $\psi$ can be further categorized into two cases: explicit constraints and hidden constraints.

**Explicit constraints with complete dual ascent** Explicit functions such as the Frobenius norm, the logarithm of the determinant, etc., are good choices for $\psi$. An efficient way to solve this constrained optimization is to solve its dual problem (mostly refers to Lagrange dual [32]). Choosing $\psi(\cdot)$ as $\log\det(\cdot)$, the detailed solution process is outlined in Appendix C. And the update formulas for $\mathbf{B}_i$ and multiplier $\lambda_i$ (dual variable) are given by

$$\mathbf{B}_i^{k+1} = \frac{\mathbf{G}_i^{-1}\left(\mathbf{\Sigma}^i + \boldsymbol{\mu}^i\left(\boldsymbol{\mu}^i\right)^T\right)\mathbf{G}_i^{-1}}{1 + 2\lambda_i^k}, \tag{23}$$

$$\lambda_i^{k+1} = \lambda_i^k + \alpha_i^k(\log\det\mathbf{B}_i^k - \log\det\mathbf{B}), \tag{24}$$

in which $\alpha_i^k$ represents the step size in the $k$-th iteration for updating the multiplier $\lambda_i$ $(i = 1, \cdots, g)$. Convergence is only guaranteed if the step size satisfies $\sum_{k=1}^{\infty}\alpha_i^k = \infty$ and $\sum_{k=1}^{\infty}(\alpha_i^k)^2 < \infty$ [32]. In our experiment, we choose a diminishing step size $1/k$ to ensure the convergence of the algorithm. The procedure, using dual ascent to diversify $\mathbf{B}_i$, is summarized in Algorithm 2 in Appendix C.

**Hidden constraints with one-step dual ascent** The weak correlated function $\psi$ can also be chosen as hidden constraint without an explicit expression. Specifically, the solution to sub-problem (22) equipped with hidden constraints $\psi$ corresponds exactly to one-step dual ascent in (23)(24). We summarize the proposition as follows:

**Proposition 3.1.** *Define an explicit weak constraint function* $\zeta : \mathbb{R}^{n^2} \to \mathbb{R}$. *For the constrained optimization problem:*

$$\begin{aligned}
\min_{\mathbf{B}_i} \quad & Q(\{\mathbf{B}_i\}_{i=1}^g, \{\mathbf{G}_i\}_{i=1}^g) \\
\mathrm{s.\,t.} \quad & \zeta(\mathbf{B}_i) = \zeta(\mathbf{B}), \quad \forall i = 1, \cdots, g,
\end{aligned}$$

the stationary point $(\{\mathbf{B}_i^{k+1}\}_{i=1}^g, \{\lambda_i^k\}_{i=1}^g)$ of the Lagrange function under given multipliers $\{\lambda_i^k\}_{i=1}^g$ satisfies:

$$\nabla_{\mathbf{B}_i} Q(\{\mathbf{B}_i^{k+1}\}_{i=1}^g, \{\mathbf{G}_i\}_{i=1}^g) - \lambda_i^k \nabla \zeta(\mathbf{B}_i^{k+1}) = 0.$$

Then there exists a constrained optimization problem with hidden weak constraint $\psi : \mathbb{R}^{n^2} \to \mathbb{R}$:

$$\min_{\mathbf{B}_i} \quad Q(\{\mathbf{B}_i\}_{i=1}^g, \{\mathbf{G}_i\}_{i=1}^g)$$
$$\text{s.t.} \quad \psi(\mathbf{B}_i) = \psi(\mathbf{B}), \quad \forall i = 1, \cdots, g,$$

such that $(\{\mathbf{B}_i^{k+1}\}_{i=1}^g, \{\lambda_i^k\}_{i=1}^g)$ is a KKT pair of the above optimization problem.

Compared to explicit formulation, hidden weak constraints, while ensuring diversification on correlation, significantly accelerate the algorithm's speed by requiring only one-step dual ascent for updating. Here, we set $\zeta(\cdot)$ as $\log \det(\cdot)$, actually solving the optimization problem under corresponding hidden constraint $\psi$. The comparison of computation time between explicit and hidden constraints, proof of Proposition 3.1 and further explanations on hidden constraints are provided in Appendix D.

Considering that it's sufficient to model elements of a block as a first order Auto-Regression (AR) process [24] in which the intra-block correlation matrix is a Toeplitz matrix, we employ this strategy for $\mathbf{B}_i$. After estimating $\mathbf{B}_i$ by dual ascent, we then apply Toeplitz correction to $\mathbf{B}_i$ as

$$\mathbf{B}_i = \text{Toeplitz}\left(\left[1, r, \cdots, r^{L-1}\right]\right)$$
$$= \begin{bmatrix} 1 & r & \cdots & r^{L-1} \\ \vdots & & & \vdots \\ r^{L-1} & r^{L-2} & \cdots & 1 \end{bmatrix}, \quad (25)$$

where $r \triangleq \frac{m_1}{m_0}$ is the approximate AR coefficient, $m_0$ represents the average of elements along the main diagonal of $\mathbf{B}_i$, and $m_1$ represents the average of elements along the main sub-diagonal of $\mathbf{B}_i$.

In conclusion, the Diversified SBL (**DivSBL**) algorithm is summarized as Algorithm 1 below.

---

**Algorithm 1** DivSBL Algorithm

---

1: **Input:** Measurement matrix $\mathbf{\Phi}$, response $\mathbf{y}$, initialized variance $\boldsymbol{\gamma}$, prior's covariance $\mathbf{\Sigma}_0$, noise's variance $\beta$, and multipliers $\boldsymbol{\lambda}^0$.          // Refer to Appendix L for initialization sensitivity.
2: **Output:** Posterior mean $\hat{\mathbf{x}}^{MAP}$, posterior covariance $\hat{\mathbf{\Sigma}}$, variance $\hat{\boldsymbol{\gamma}}$, correlation $\hat{\mathbf{B}}_i$, noise $\hat{\beta}$.
3: **repeat**
4:    **if** $\text{mean}(\boldsymbol{\gamma}_{l.}) < $ threshold **then**
5:       Prune $\boldsymbol{\gamma}_{l.}$ from the model (set $\boldsymbol{\gamma}_{l.} = \mathbf{0}$).       // Zero out small energy for efficiency.
6:       Set the corresponding $\boldsymbol{\mu}^l = \mathbf{0}, \mathbf{\Sigma}^l = \mathbf{0}_{L \times L}$.
7:    **end if**
8:    Update $\gamma_{ij}$ by (18).       // Update diversified variance.
9:    Update $\mathbf{B}$ by (21).       // Avoid overfitting.
10:    Update $\mathbf{B}_i, \lambda_i$ by (23)(24).       // Diversified correlation.
11:    Execute Toeplitz correction for $\mathbf{B}_i$ using (25).[3]
12:    Update $\boldsymbol{\mu}$ and $\mathbf{\Sigma}$ by (13)(14).
13:    Update $\beta$ using (20).
14: **until** convergence criterion met
15: $\hat{\mathbf{x}}^{MAP} = \boldsymbol{\mu}$.       // Use posterior mean as estimate.

---

## 4 Global minimum and local minima

For the sake of simplicity, we denote the true signal as $\mathbf{x}_{\text{true}}$, which is the sparsest among all feasible solutions. The block sparsity of the true signal is denoted as $K_0$, indicating the presence of $K_0$ blocks. Let $\tilde{\mathbf{G}} \triangleq \text{diag}\left(\sqrt{\gamma_{11}}, \cdots, \sqrt{\gamma_{gL}}\right)$, $\tilde{\mathbf{B}} \triangleq \text{diag}(\mathbf{B}_1, \cdots, \mathbf{B}_g)$, thus $\mathbf{\Sigma}_0 = \tilde{\mathbf{G}}\tilde{\mathbf{B}}\tilde{\mathbf{G}}$. Additionally, we assume that the measurement matrix $\mathbf{\Phi}$ satisfies the URP condition [2]. We employed various techniques to overcome highly non-linear structure of $\boldsymbol{\gamma}$ in diversified block sparse prior (5), resulting in following global and local optimality theorems.

### 4.1 Analysis of global minimum

By introducing a negative sign to the cost function (16), we have the following result on the property of global minimum and the threshold for block sparsity $K_0$.

---

[3]The necessity of each step for updating $\mathbf{B}_i$ in line 9-11 of Algorithm 1 is detailed in Appendix A.

Table 1: Reconstruction error (NMSE) and Correlation (mean±std) for synthetic signals. Our algorithm is marked in  blue , and the best-performing metrics are displayed in **bold**.

| Algorithm | NMSE | Corr |
|---|---|---|
| **Homoscedastic** | | |
| BSBL | 0.0132±0.0069 | 0.9936±0.0034 |
| PC-SBL | 0.0450±0.0188 | 0.9784±0.0090 |
| SBL | 0.0263±0.0129 | 0.9825±0.0062 |
| Group Lasso | 0.0215±**0.0052** | 0.9925±**0.0020** |
| Group BPDN | 0.0378±0.0087 | 0.9812±0.0044 |
| StructOMP | 0.0508±0.0157 | 0.9760±0.0073 |
| DivSBL | **0.0094**±0.0053 | **0.9955**±0.0026 |
| **Heteroscedastic** | | |
| BSBL | 0.0245±0.0125 | 0.9883±0.0047 |
| PC-SBL | 0.0421±0.0169 | 0.9798±0.0082 |
| SBL | 0.0274±0.0095 | 0.9873±0.0040 |
| Group Lasso | 0.0806±0.0180 | 0.9642±0.0096 |
| Group BPDN | 0.0857±0.0173 | 0.9608±0.0096 |
| StructOMP | 0.0419±0.0123 | 0.9803±0.0061 |
| DivSBL | **0.0086**±0.0041 | **0.9958**±0.0020 |

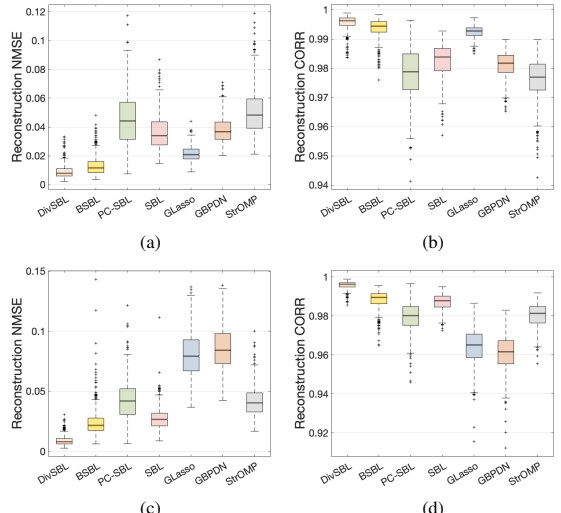

Figure 3: The consistency of multiple experiments with homoscedastic signals for (a) NMSE (b) Correlation, and with heteroscedastic signals for (c) NMSE and (d) Correlation.

**Theorem 4.1.** *As $\beta \to \infty$ and $K_0 < (M+1)/2L$, the unique global minimum $\widehat{\boldsymbol{\gamma}} \triangleq (\widehat{\gamma}_{11}, \ldots, \widehat{\gamma}_{gL})^T$ yields a recovery $\hat{\mathbf{x}}$ by (13) that is equal to $\mathbf{x}_{true}$, regardless of the estimated $\widehat{\mathbf{B}}_i$ ($\forall i$).*

The proof draws inspiration from [23] and is detailed in Appendix E. Theorem 4.1 shows that, in noiseless case, achieving the global minimum of variance enables exact recovery of the true signal, provided the block sparsity of the signal adheres to the given upper bound.

### 4.2 Analysis of local minima

We provide two lemmas firstly. The proofs are detailed in Appendices F and G.

**Lemma 4.2.** *For any semi-definite positive symmetric matrix $\mathbf{Z} \in \mathbb{R}^{M \times M}$, the constraint $\mathbf{Z} \succeq \mathbf{\Phi}\mathbf{\Sigma}_0\mathbf{\Phi}^T + \beta^{-1}\mathbf{I}$ is convex with respect to $\mathbf{Z}$ and $\left(\sqrt{\boldsymbol{\gamma}} \otimes \sqrt{\boldsymbol{\gamma}}\right)$.*

**Lemma 4.3.** $\mathbf{y}^T \mathbf{\Sigma}_y^{-1} \mathbf{y} = C \Leftrightarrow \mathbf{P}\left(\sqrt{\boldsymbol{\gamma}} \otimes \sqrt{\boldsymbol{\gamma}}\right) = \mathbf{b}$ *for any constant $C$, where $\mathbf{b} \triangleq \mathbf{y} - \beta^{-1}\mathbf{u}$, $\mathbf{P} \triangleq \left[(\mathbf{u}^T\mathbf{\Phi}) \otimes \mathbf{\Phi}\right] \operatorname{diag}\left(vec(\tilde{\mathbf{B}})\right)$, and $\mathbf{u}$ is a vector satisfying $\mathbf{y}^T\mathbf{u} = C$.*

It's clear that $\mathbf{P}$ is a full row rank matrix, i.e, $r(\mathbf{P}) = M$. Given the above lemmas, we arrive at the following result, which is proven in Appendix H.

**Theorem 4.4.** *Every local minimum of the cost function* (16) *with respect to $\boldsymbol{\gamma}$ satisfies $||\hat{\boldsymbol{\gamma}}||_0 \leq \sqrt{M}$, irrespective of the estimated $\widehat{\mathbf{B}}_i$ ($\forall i$) and $\beta$.*

Theorem 4.4 establishes an upper bound on the sparsity level of any local minimum for the cost function in terms of the parameter $\boldsymbol{\gamma}$. Therefore, together with Theorem 4.1, these results ensure the sparsity of the final solution obtained.

## 5 Experiments

In this section, we compare DivSBL with the following six algorithms:[4] 1. Block-based algorithms: (1) BSBL, (2) Group Lasso, (3) Group BPDN. 2. Pattern-based algorithms: (4) PC-SBL, (5) StructOMP. 3. Sparse learning (without structural information): (6) SBL. Results are averaged over 100 or 500 random runs (based on computational scale), with SNR ranging from 15-25 dB

---

[4]Matlab codes for our algorithm are available at `https://github.com/YanhaoZhang1/DivSBL` .

except the test for varied noise levels. 'Normalized Mean Squared Error (NMSE)', defined as $||\hat{x} - x_{\text{true}}||_2^2/||x_{\text{true}}||_2^2$, and 'Correlation (Corr)' (cosine similarity) are used to compare algorithms. [5]

## 5.1 Synthetic signal data

We initially test on synthetic signal data, including homoscedastic (provided by [24]) and heteroscedastic data, where block size, location, non-zero quantity, and signal variance are randomly generated, mimicking real-world data patterns. The reconstruction results are provide in Appendix I.1. Table 1 shows that DivSBL achieves the lowest NMSE and the highest Correlation on both scenarios. To more intuitively demonstrate the statistically significant improvements of the conclusion, we provide box plots of the experimental results on both homoscedastic and heteroscedastic data in Figure 3 .

Unlike many frequentist approaches that require more complex debiasing methods to construct confidence intervals, the Bayesian approach offers a straightforward way to obtain credible intervals for point estimates. For more results related to Bayesian methods, please refer to Appendix I.2.

## 5.2 The robustness of pre-defined block sizes

As mentioned in Section 1, block-based algorithms require presetting block sizes, and their performance is sensitive to these parameters, posing challenges in practice. This experiment assesses the robustness of block-based algorithms with predefined block sizes. The test setup is shown in Figure 4. We vary preset block sizes and conduct 100 experiments for all algorithms. Confidence intervals in Figure 4 depict reconstruction error for statistical significance.

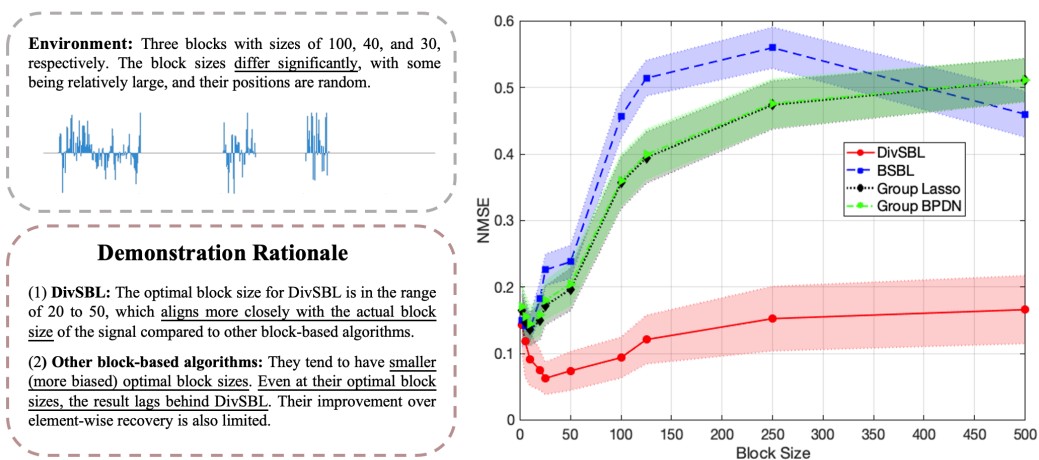

Figure 4: NMSE variation with changing preset block sizes.

**Resolves the longstanding sensitivity issue of block-based algorithms.** DivSBL demonstrates strong robustness to the preset block sizes, effectively addressing the sensitivity issue that block-based algorithms commonly encounter with respect to block sizes.

Figure 5 visualizes the posterior variance learning on the signal to demonstrate DivSBL's ability to adaptively identify the true blocks. The algorithms are tested with preset block sizes of 20 (small), 50 (medium), and 125 (large), respectively, to show how each algorithm learns the blocks when block structure is misspecified. As expected in Section 2.1 and Figure 2 , DivSBL is able to adaptively find the true block through diversification learning and remains robust to the preset block size.

**Exhibits enhanced recovery capability in challenging scenarios.** The optimal block size for DivSBL is around 20-50, which is more consistent with the true block sizes. This indicates that when true block sizes are large and varied, DivSBL can effectively capture richer information within each block by setting larger block sizes, thereby significantly improving the recovery performance. In contrast, other algorithms do not perform as well as DivSBL, even at their optimal block sizes.

---

[5]NMSE quantifies the numerical closeness of two vectors, while correlation measures the accuracy of estimating a target support set and the level of recovery similarity within that set (also utilized in [33]). Therefore, employing both metrics together can more comprehensively reflect the structural sparse recovery of the target.

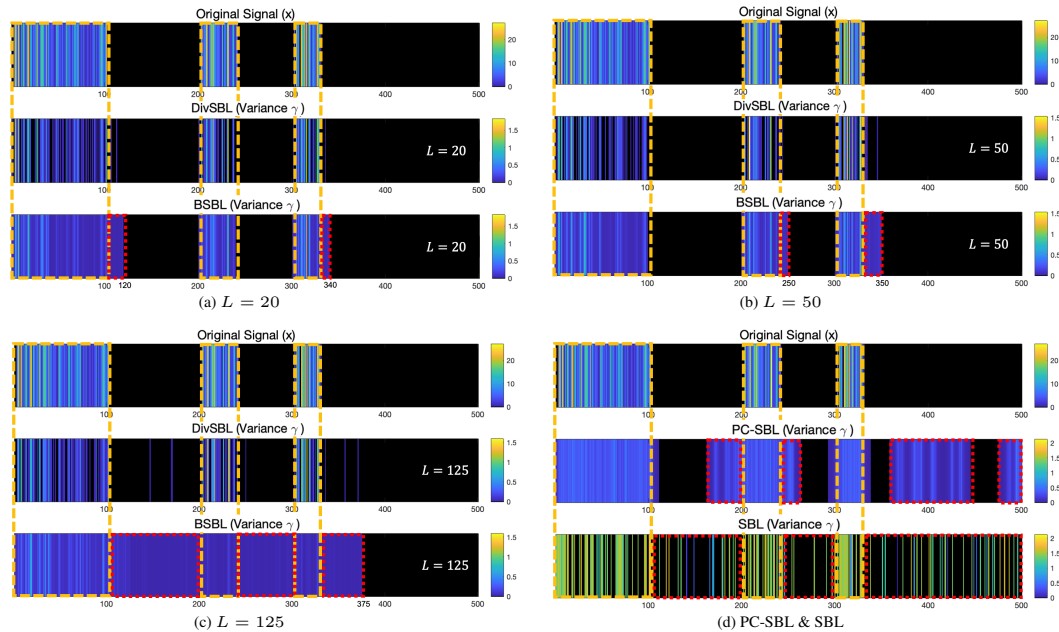

Figure 5: Variance learning

Table 2: Reconstruction errors (NMSE $\pm$ std) under different noise levels (sample rate=0.25).

| SNR | BSBL | PC-SBL | SBL | Group BPDN | Group Lasso | StructOMP | DivSBL |
|---|---|---|---|---|---|---|---|
| 10 | $0.235 \pm 0.052$ | $0.283 \pm \mathbf{0.049}$ | $0.391 \pm 0.064$ | $0.223 \pm 0.055$ | $0.215 \pm 0.055$ | $0.437 \pm 0.129$ | $\mathbf{0.191 \pm 0.076}$ |
| 15 | $0.100 \pm 0.022$ | $0.173 \pm 0.039$ | $0.235 \pm 0.031$ | $0.149 \pm 0.058$ | $0.139 \pm 0.057$ | $0.167 \pm 0.043$ | $\mathbf{0.074 \pm 0.016}$ |
| 20 | $0.046 \pm 0.010$ | $0.107 \pm 0.028$ | $0.180 \pm 0.018$ | $0.122 \pm 0.062$ | $0.111 \pm 0.062$ | $0.067 \pm 0.020$ | $\mathbf{0.035 \pm 0.010}$ |
| 25 | $0.025 \pm 0.006$ | $0.068 \pm 0.027$ | $0.167 \pm 0.018$ | $0.113 \pm 0.057$ | $0.102 \pm 0.057$ | $0.030 \pm 0.012$ | $\mathbf{0.019 \pm 0.005}$ |
| 30 | $0.015 \pm 0.006$ | $0.046 \pm 0.023$ | $0.160 \pm 0.016$ | $0.103 \pm 0.054$ | $0.093 \pm 0.053$ | $0.019 \pm 0.011$ | $\mathbf{0.010 \pm 0.004}$ |
| 35 | $0.011 \pm 0.004$ | $0.032 \pm 0.017$ | $0.155 \pm 0.019$ | $0.100 \pm 0.070$ | $0.088 \pm 0.070$ | $0.013 \pm 0.008$ | $\mathbf{0.009 \pm 0.003}$ |
| 40 | $0.009 \pm 0.004$ | $0.027 \pm 0.020$ | $0.155 \pm 0.015$ | $0.095 \pm 0.061$ | $0.084 \pm 0.060$ | $0.010 \pm 0.006$ | $\mathbf{0.007 \pm 0.002}$ |
| 45 | $0.008 \pm 0.005$ | $0.025 \pm 0.016$ | $0.153 \pm 0.015$ | $0.099 \pm 0.053$ | $0.087 \pm 0.052$ | $0.011 \pm 0.010$ | $\mathbf{0.006 \pm 0.002}$ |
| 50 | $0.008 \pm 0.004$ | $0.024 \pm 0.017$ | $0.155 \pm 0.015$ | $0.101 \pm 0.063$ | $0.090 \pm 0.062$ | $0.009 \pm 0.006$ | $\mathbf{0.007 \pm 0.003}$ |

### 5.3 1D audioSet

As shown in Figure 14, audio signals exhibit block sparse structures in discrete cosine transform (DCT) basis, which is well-suited for assessing block sparse algorithms. In this subsection, we carry out experiments on real-world audios, which are randomly chosen in *AudioSet* [34]. The reconstruction results are present in Appendix J. In the main text, we focus on analyzing DivSBL's sensitivity to sample rate, evaluating its performance across different noise levels, and investigating its phase transition properties.

**The sensitivity of sample rate** The algorithms are tested on audio sets to investigate the sensitivity of sample rate ($M/N$) varied from 0.25 to 0.55. The result is visualized in Figure 16. Notably, DivSBL emerges as the top performer across diverse sampling rates, showing a consistent 1 dB enhancement in NMSE compared to the best-performing algorithm among others.

**The performance under various noise levels** We assess each algorithm as Signal-to-Noise Ratio (SNR) varied from 10 to 50 and include the standard deviation from 100 random experiments on audio sets. Here, we present the performance under the minimum sampling rate 0.25 tested before, which represents a challenging recovery scenario. As shown in Table 2, the performance of all algorithms improves with higher SNR. Notably, DivSBL consistently leads across all SNR levels.

**Phase Transition** This audio data contains approximately 90 non-zero elements in DCT domain ($K = 90$), which constitutes about 20% of the total dimensionality ($N = 480$). Therefore, we start the test with a sampling rate of a same 20%. In this scenario, $M/K$ is roughly 1 and increases with the sampling rate. Concurrently, the signal-to-noise ratio (SNR) varies gradually from 10 to 50.

The phase transition diagram in Figure 6 shows that DivSBL performs well at more extreme sampling rates and is better suited for lower SNR conditions.

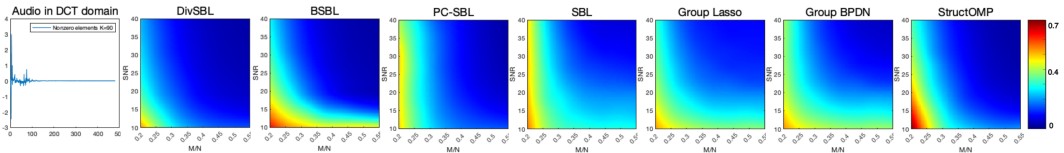

Figure 6: Phase transition diagram under different SNR and measurements.

## 5.4 2D image reconstruction

In 2D image experiments, we utilize a standard set of grayscale images compiled from two sources [6]. As depicted in Figure 17, the images exhibit block sparsity in discrete wavelet domain. In Section 5.3, we've shown DivSBL's leading performance across diverse sampling rates. Here, we use a 0.5 sample rate and the reconstruction errors are in Table 3 and Appendix K. DivSBL's reconstructions surpass others, with an average improvement of **9.8%** on various images.

Table 3: Reconstructed error (Square root of NMSE $\pm$ std) of the test images.

| Algorithm | Parrot | Cameraman | Lena | Boat | House | Barbara | Monarch | Foreman |
|---|---|---|---|---|---|---|---|---|
| BSBL | $0.139 \pm \mathbf{0.004}$ | $0.156 \pm 0.006$ | $0.137 \pm \mathbf{0.004}$ | $0.179 \pm \mathbf{0.007}$ | $0.146 \pm 0.007$ | $0.142 \pm \mathbf{0.004}$ | $0.272 \pm 0.009$ | $0.125 \pm 0.007$ |
| PC-SBL | $0.133 \pm 0.013$ | $0.150 \pm 0.012$ | $0.134 \pm 0.013$ | $0.159 \pm 0.014$ | $0.137 \pm 0.013$ | $0.137 \pm 0.013$ | $0.208 \pm 0.010$ | $0.126 \pm 0.014$ |
| SBL | $0.225 \pm 0.121$ | $0.247 \pm 0.141$ | $0.223 \pm 0.129$ | $0.260 \pm 0.114$ | $0.238 \pm 0.125$ | $0.228 \pm 0.119$ | $0.458 \pm 0.106$ | $0.175 \pm 0.099$ |
| GLasso | $0.139 \pm 0.017$ | $0.153 \pm 0.016$ | $0.134 \pm 0.017$ | $0.159 \pm 0.018$ | $0.141 \pm 0.018$ | $0.135 \pm 0.016$ | $0.216 \pm 0.020$ | $0.124 \pm 0.017$ |
| GBPDN | $0.138 \pm 0.017$ | $0.153 \pm 0.017$ | $0.134 \pm 0.017$ | $0.159 \pm 0.019$ | $0.133 \pm 0.019$ | $0.135 \pm 0.017$ | $0.218 \pm 0.022$ | $0.123 \pm 0.017$ |
| StrOMP | $0.161 \pm 0.014$ | $0.184 \pm 0.013$ | $0.159 \pm 0.013$ | $0.187 \pm 0.014$ | $0.162 \pm 0.014$ | $0.164 \pm 0.013$ | $0.248 \pm 0.015$ | $0.149 \pm 0.016$ |
| **DivSBL** | $\mathbf{0.117} \pm 0.007$ | $\mathbf{0.142} \pm \mathbf{0.006}$ | $\mathbf{0.114} \pm 0.005$ | $\mathbf{0.150} \pm 0.008$ | $\mathbf{0.120} \pm \mathbf{0.006}$ | $\mathbf{0.120} \pm 0.005$ | $\mathbf{0.203} \pm \mathbf{0.008}$ | $\mathbf{0.101} \pm \mathbf{0.007}$ |

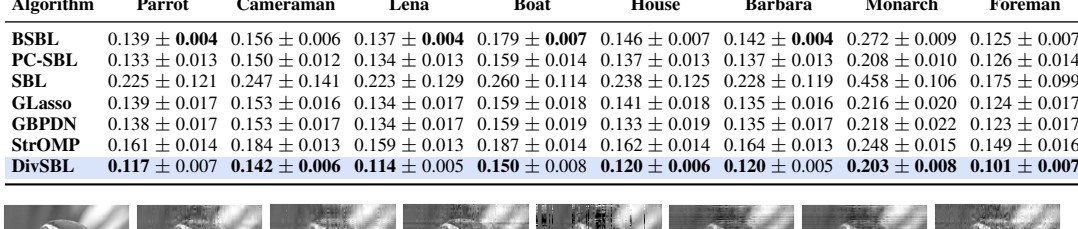
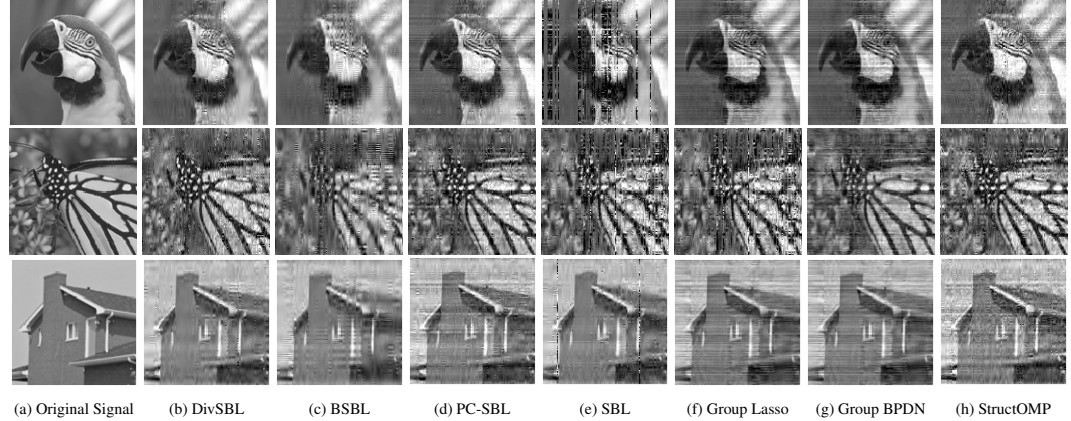

(a) Original Signal  (b) DivSBL  (c) BSBL  (d) PC-SBL  (e) SBL  (f) Group Lasso  (g) Group BPDN  (h) StructOMP

Figure 7: Reconstruction results for Parrot, Monarch and House images.

In Figure 7, we display the final reconstructions of Parrot, Monarch and House images as examples. DivSBL is capable of preserving the finer features of the parrot, such as cheek, eye, etc., and recovering the background smoothly with minimal error stripes. As for Monarch and House images, nearly every reconstruction introduces undesirable artifacts and stripes, while images restored by DivSBL show the least amount of noise patterns, demonstrating the most effective restoration.

## 6 Conclusions

This paper established a new Bayesian learning model by introducing diversified block sparse prior, to effectively capture the prevalent block sparsity observed in real-world data. The novel Bayesian model effectively solved the sensitivity issue in existing block sparse learning methods, allowing for adaptive block estimation and reducing the risk of overfitting. The proposed algorithm DivSBL, based on this model, enjoyed solid theoretical guarantees on both convergence and sparsity theory. Experimental results demonstrated its state-of-the-art performance on multimodal data. Future works include exploration on more effective weak constraints for correlation matrices, and applications on supervised learning tasks such as regression and classification.

---

[6]Available at http://dsp.rice.edu/software/DAMP-toolbox and http://see.xidian.edu.cn/faculty/wsdong/NLR_Exps.htm

## Acknowledgments and Disclosure of Funding

This research was supported by National Key R&D Program of China under grant 2021YFA1003300.

The authors would like to thank all the reviewers for their helpful comments. Their suggestions have helped us enhance our experiments to present a more comprehensive demonstration of the effectiveness of DivSBL.

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

## A Experimental result of diversifying correlation matrices

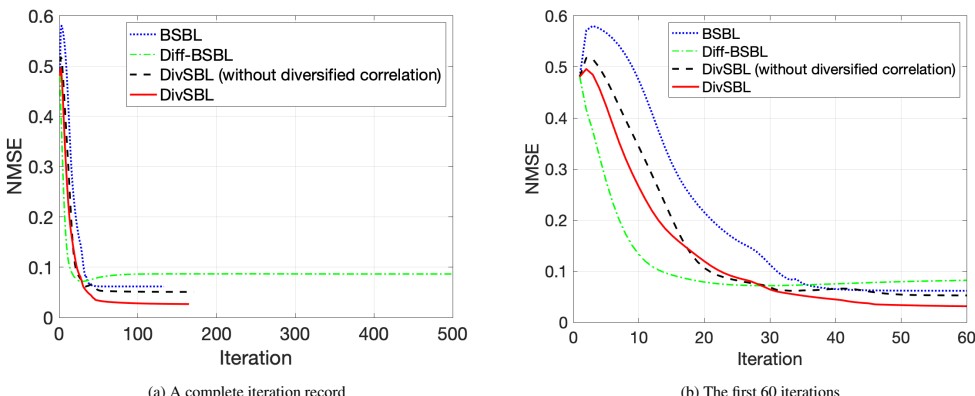

(a) A complete iteration record

(b) The first 60 iterations

Figure 8: NMSE with iteration number.

Below, we will explain the necessity of the three steps for updating $\mathbf{B}$ in DivSBL:

(1) Firstly, the purpose of the first step, estimating $\mathbf{B}$ under a strong constraint, is to avoid overfitting. As shown by the green line (Diff-BSBL) in Figure 8 , algorithm without strong constraints tend to worsen NMSE after several iterations due to overfitting.

(2) Since the strong constraint in (1) forces all blocks to have the same correlation, it leads to the loss of specificity in correlation matrices within different blocks. Therefore, the motivation for the second step, applying weak constraints, is to make the correlations within blocks similar to some extent while preserving their individual specificities. As the black line in Figure 8 (DivSBL-without diversified correlation) shows, DivSBL without weak constraints fails to capture the specificity within blocks, resulting in a loss of accuracy and slower speed compare to DivSBL.

(3) The third step, Toeplitzization, inherits the advantages of BSBL, allowing the correlation matrices to have a reasonable structure.

Overall, while BSBL utilizes a single variance and a shared correlation matrix within each block, DivSBL incorporates diversification into both variance and correlation matrix modeling, making it more flexible and enhancing its performance.

## B Derivation of hyperparameters updating formulas

In this paper, lowercase and uppercase bold symbols are employed to represent vectors and matrices, respectively. $\det(\mathbf{A})$ denotes the determinant of matrix $\mathbf{A}$. $\mathrm{tr}(\mathbf{A})$ means the trace of $\mathbf{A}$. Matrix $\mathrm{diag}(\mathbf{A}_1, \ldots, \mathbf{A}_g)$ represents the block diagonal matrix with the matrices $\{\mathbf{A}_i\}_{i=1}^g$ placed along the main diagonal, and $\mathrm{Diag}(\mathbf{A})$ represents the extraction of the diagonal elements from matrix $\mathbf{A}$ to create a vector. We observe that

$$Q(\{\mathbf{G}_i\}_{i=1}^g, \{\mathbf{B}_i\}_{i=1}^g) \propto -\frac{1}{2} E_{x|y;\Theta^{t-1}}(\log|\boldsymbol{\Sigma}_0| + \mathbf{x}^T \boldsymbol{\Sigma}_0 \mathbf{x})$$

$$= -\frac{1}{2}\log|\boldsymbol{\Sigma}_0| - \frac{1}{2}\mathrm{tr}\left[\boldsymbol{\Sigma}_0^{-1}(\boldsymbol{\Sigma} + \boldsymbol{\mu}\boldsymbol{\mu}^T)\right], \tag{26}$$

in which $\boldsymbol{\Sigma}_0$ can be reformulated as

$$\boldsymbol{\Sigma}_0 = \mathrm{diag}\left\{\mathbf{G}_1\mathbf{B}_1\mathbf{G}_1, \cdots, \mathbf{G}_g\mathbf{B}_g\mathbf{G}_g\right\} = \mathbf{D}_{-i} + \begin{pmatrix} \mathbf{0} \\ \mathbf{I}_L \\ \mathbf{0} \end{pmatrix} \mathbf{G}_i\mathbf{B}_i\mathbf{G}_i \begin{pmatrix} \mathbf{0} & \mathbf{I}_L & \mathbf{0} \end{pmatrix}$$

$$= \mathbf{D}_{-i} + \begin{pmatrix} \mathbf{0} \\ \mathbf{I}_L \\ \mathbf{0} \end{pmatrix} (\sqrt{\gamma_{ij}}\mathbf{P}_j + \mathbf{W}_{-j}^i)\mathbf{B}_i(\sqrt{\gamma_{ij}}\mathbf{P}_j + \mathbf{W}_{-j}^i)\begin{pmatrix} \mathbf{0} & \mathbf{I}_L & \mathbf{0} \end{pmatrix}, \tag{27}$$

where

$$\mathbf{D}_{-i} = \text{diag}\left\{\mathbf{G}_1\mathbf{B}_1\mathbf{G}_1, \cdots, \mathbf{G}_{i-1}\mathbf{B}_{i-1}\mathbf{G}_{i-1}, \mathbf{0}_{L\times L}, \mathbf{G}_{i+1}\mathbf{B}_{i+1}\mathbf{G}_{i+1}, \cdots, \mathbf{G}_g\mathbf{B}_g\mathbf{G}_g\right\},$$

$$\mathbf{P}_j = \text{diag}\{\delta_{1j}, \delta_{2j}, \cdots, \delta_{Lj}\},$$

$$\mathbf{W}^i_{-j} = \text{diag}\{\sqrt{\gamma_{i1}}, \cdots, \sqrt{\gamma_{i,j-1}}, 0, \sqrt{\gamma_{i,j+1}}, \cdots, \sqrt{\gamma_{iL}}\},$$

and the Kronecker delta function here, denoted by $\delta_{ij}$, is defined as

$$\delta_{ij} = \begin{cases} 1, & \text{if } i = j, \\ 0, & \text{otherwise.} \end{cases}$$

Hence, $\mathbf{\Sigma}_0$ is split into two components in (27), with the second term of the summation only depending on $\mathbf{G}_i$ and $\mathbf{B}_i$, and the first part $\mathbf{D}_{-i}$ being entirely unrelated to them. Then, we can update each $\mathbf{B}_i$ and $\mathbf{G}_i$ independently, allowing us to learn diverse $\mathbf{B}_i$ and $\mathbf{G}_i$ for different blocks.

The gradient of (26) with respect to $\sqrt{\gamma_{ij}}$ can be expressed as

$$\frac{\partial Q(\{\mathbf{G}_i\}_{i=1}^g, \{\mathbf{B}_i\}_{i=1}^g)}{\partial\sqrt{\gamma_{ij}}} = \frac{\partial(-\frac{1}{2}\log|\mathbf{\Sigma}_0|)}{\partial\sqrt{\gamma_{ij}}} + \frac{\partial(-\frac{1}{2}\text{tr}\left[\mathbf{\Sigma}_0^{-1}(\mathbf{\Sigma} + \boldsymbol{\mu}\boldsymbol{\mu}^T)\right])}{\partial\sqrt{\gamma_{ij}}}.$$

The first term results in

$$\frac{\partial(-\frac{1}{2}\log|\mathbf{\Sigma}_0|)}{\partial\sqrt{\gamma_{ij}}} = -\text{tr}\left(\mathbf{P}_j(\mathbf{G}_i\mathbf{B}_i\mathbf{G}_i)^{-1}\mathbf{G}_i\mathbf{B}_i\right) = -\frac{1}{\sqrt{\gamma_{ij}}},$$

and the second term yields

$$\frac{\partial(-\frac{1}{2}\text{tr}\left[\mathbf{\Sigma}_0^{-1}(\mathbf{\Sigma} + \boldsymbol{\mu}\boldsymbol{\mu}^T)\right])}{\partial\sqrt{\gamma_{ij}}} = \text{tr}\left[\mathbf{P}_j(\mathbf{G}_i\mathbf{B}_i\mathbf{G}_i)^{-1}(\mathbf{\Sigma}^i + \boldsymbol{\mu}^i(\boldsymbol{\mu}^i)^T)\mathbf{G}_i^{-1}\right]$$

$$= \text{tr}\left[\mathbf{B}_i^{-1}\mathbf{G}_i^{-1}(\mathbf{\Sigma}^i + \boldsymbol{\mu}^i(\boldsymbol{\mu}^i)^T)\mathbf{P}_j\gamma_{ij}^{-1}\right],$$

where $\boldsymbol{\mu}^i \in \mathbb{R}^{L\times 1}$ represents the $i$-th block in $\boldsymbol{\mu}$, and $\mathbf{\Sigma}^i \in \mathbb{R}^{L\times L}$ denotes the $i$-th block in $\mathbf{\Sigma}$ [7]. Using $\mathbf{A}_1(\mathbf{M} + \mathbf{N})\mathbf{A}_2 = \mathbf{A}_1\mathbf{M}\mathbf{A}_2 + \mathbf{A}_1\mathbf{N}\mathbf{A}_2$, the formula above can be further transformed into

$$\frac{\partial(-\frac{1}{2}\text{tr}\left[\mathbf{\Sigma}_0^{-1}(\mathbf{\Sigma} + \boldsymbol{\mu}\boldsymbol{\mu}^T)\right])}{\partial\sqrt{\gamma_{ij}}} = \text{tr}\left[\mathbf{B}_i^{-1}\frac{1}{\sqrt{\gamma_{ij}}}\mathbf{P}_j(\mathbf{\Sigma}^i + \boldsymbol{\mu}^i(\boldsymbol{\mu}^i)^T)\mathbf{P}_j\gamma_{ij}^{-1}\right]$$

$$+ \text{tr}\left[\mathbf{B}_i^{-1}(\mathbf{I} - \mathbf{P}_j)\mathbf{G}_i^{-1}(\mathbf{\Sigma}^i + \boldsymbol{\mu}^i(\boldsymbol{\mu}^i)^T)\mathbf{P}_j\gamma_{ij}^{-1}\right]$$

$$= (\frac{1}{\sqrt{\gamma_{ij}}})^3\mathbf{A}_{ij} + \frac{1}{\gamma_{ij}}\mathbf{T}_{ij},$$

in which, $\mathbf{T}_{ij}$ and $\mathbf{A}_{ij}$ are independent of $\sqrt{\gamma_{ij}}$, and their expressions are

$$\mathbf{T}_{ij} = \left[(\mathbf{B}_i^{-1})_{j\cdot} \odot \text{diag}(\mathbf{W}^i_{-j})^{-1}\right] \cdot \left(\mathbf{\Sigma}^i + \boldsymbol{\mu}^i(\boldsymbol{\mu}^i)^T\right)_{\cdot j},$$

$$\mathbf{A}_{ij} = (\mathbf{B}_i^{-1})_{jj} \cdot \left(\mathbf{\Sigma}^i + \boldsymbol{\mu}^i\left(\boldsymbol{\mu}^i\right)^T\right)_{jj}.$$

Thus, the derivative of $Q(\Theta)$ with respect to $\sqrt{\gamma_{ij}}$ reads as

$$\frac{\partial Q(\Theta)}{\partial\sqrt{\gamma_{ij}}} = -\frac{1}{\sqrt{\gamma_{ij}}} + (\frac{1}{\sqrt{\gamma_{ij}}})^3\mathbf{A}_{ij} + \frac{1}{\gamma_{ij}}\mathbf{T}_{ij}. \tag{28}$$

It is important to note that the variance should be non-negative. So by setting (28) equal to zero, we obtain the update formulation of $\gamma_{ij}$ as

$$\gamma_{ij} = \frac{4\mathbf{A}_{ij}^2}{(\sqrt{\mathbf{T}_{ij}^2 + 4\mathbf{A}_{ij}} - \mathbf{T}_{ij})^2}. \tag{29}$$

As for the gradient of (26) with respect to $\mathbf{B}_i$, we have

$$\frac{\partial Q(\{\mathbf{G}_i\}_{i=1}^g, \{\mathbf{B}_i\}_{i=1}^g)}{\partial\mathbf{B}_i} = -\frac{1}{2}\mathbf{B}_i^{-1} + \frac{1}{2}\mathbf{B}_i^{-1}\mathbf{G}_i^{-1}(\mathbf{\Sigma}^i + \boldsymbol{\mu}^i(\boldsymbol{\mu}^i)^T)\mathbf{G}_i^{-1}\mathbf{B}_i^{-1}.$$

Setting it equal to zero, the learning rule for $\mathbf{B}_i$ is given by

$$\mathbf{B}_i = \mathbf{G}_i^{-1}\left(\mathbf{\Sigma}^i + \boldsymbol{\mu}^i\left(\boldsymbol{\mu}^i\right)^T\right)\mathbf{G}_i^{-1}. \tag{30}$$

---

[7]Using MATLAB notation, $\boldsymbol{\mu}^i \triangleq \boldsymbol{\mu}((i-1)L + 1 : iL)$, $\mathbf{\Sigma}^i \triangleq \mathbf{\Sigma}((i-1)L + 1 : iL, (i-1)L + 1 : iL)$.

## C The procedure for solving the constrained optimization problem

To clarify the dual problem of (P), we firstly express (P)'s Lagrange function as

$$\mathcal{L}(\{\mathbf{B}_i\}_{i=1}^g; \{\lambda_i\}_{i=1}^g) = \frac{1}{2}\log\det\boldsymbol{\Sigma}_0 + \frac{1}{2}\operatorname{tr}\left[\boldsymbol{\Sigma}_0^{-1}(\boldsymbol{\Sigma} + \boldsymbol{\mu}\boldsymbol{\mu}^T)\right] + \sum_{i=1}^g \lambda_i(\log\det\mathbf{B}_i - \log\det\mathbf{B}). \tag{31}$$

Since the constraints in (P) are equalities, we do not impose any requirements on the multipliers $\{\lambda_i\}_{i=1}^g$. The primal and dual problems in terms of $\mathcal{L}$ are given by

$$\min_{\{\mathbf{B}_i\}_{i=1}^g} \max_{\{\lambda_i\}_{i=1}^g} \quad \mathcal{L}(\{\mathbf{B}_i\}_{i=1}^g; \{\lambda_i\}_{i=1}^g), \tag{P}$$

$$\max_{\{\lambda_i\}_{i=1}^g} \min_{\{\mathbf{B}_i\}_{i=1}^g} \quad \mathcal{L}(\{\mathbf{B}_i\}_{i=1}^g; \{\lambda_i\}_{i=1}^g), \tag{D}$$

respectively. Although the objective here is non-convex, dual ascent method takes advantage of the fact that the dual problem is always convex [32], and we can get a lower bound of (P) by solving (D). Specifically, dual ascent method employs gradient ascent on the dual variables. As long as the step sizes are chosen properly, the algorithm would converge to a local maximum. We will demonstrate how to choose the step sizes in the following paragraph.

According to this framework, we first solve the inner minimization problem of the dual problem (D). Keeping the multipliers $\{\lambda_i\}_{i=1}^g$ fixed, the inner problem is

$$\mathbf{B}_i^{k+1} \in \arg\min_{\mathbf{B}_i} \mathcal{L}(\{\mathbf{B}_i\}_{i=1}^g; \{\lambda_i^k\}_{i=1}^g) = \arg\min_{\mathbf{B}_i} \mathcal{L}(\mathbf{B}_i; \lambda_i^k), \tag{32}$$

where the superscript $k$ implies the $k$-th iteration. According to the first-order optimality condition, the primal solution for (32) is as follows:

$$\mathbf{B}_i^{k+1} = \frac{\mathbf{G}_i^{-1}\left(\boldsymbol{\Sigma}^i + \boldsymbol{\mu}^i\left(\boldsymbol{\mu}^i\right)^T\right)\mathbf{G}_i^{-1}}{1 + 2\lambda_i^k}. \tag{33}$$

Subsequently, the outer maximization problem for the multiplier $\lambda_i$ (dual variable) can be addressed using the gradient ascent method. The update formulation is obtained by

$$\begin{aligned}
\lambda_i^{k+1} &= \lambda_i^k + \alpha_i^k \nabla_{\lambda_i}\mathcal{L}(\{\mathbf{B}_i^k\}_{i=1}^g; \{\lambda_i\}_{i=1}^g) \\
&= \lambda_i^k + \alpha_i^k \nabla_{\lambda_i}\mathcal{L}(\mathbf{B}_i^k; \lambda_i) \\
&= \lambda_i^k + \alpha_i^k(\log\det\mathbf{B}_i^k - \log\det\mathbf{B}),
\end{aligned} \tag{34}$$

in which, $\alpha_i^k$ represents the step size in the $k$-th iteration for updating the multiplier $\lambda_i$ ($i = 1...g$). Convergence is only guaranteed if the step size satisfies $\sum_{k=1}^{\infty}\alpha_i^k = \infty$ and $\sum_{k=1}^{\infty}(\alpha_i^k)^2 < \infty$ [32]. Therefore, we choose a diminishing step size $1/k$ to ensure the convergence. The procedure, using dual ascent method to diversify $\mathbf{B}_i$, is summarized in Algorithm 2 as follows:

---
**Algorithm 2** Diversifying $\mathbf{B}_i$

---
1: **Input:** Initialized $\lambda_i^0 \in \mathbf{R}, \varepsilon > 0$, the common intra-block correlation $\mathbf{B}$ obtained from (21).
2: **Output:** $\mathbf{B}_i, i = 1, \cdots, g$.
3: Set iteration count $k = 1$.
4: **repeat**
5:     Set step size $\alpha_i^k = 1/k$.
6:     Update $\mathbf{B}_i^{k+1}$ using (33).
7:     Update $\lambda_i^{k+1}$ using (34).
8:     $k := k + 1$;
9: **until** $|\log\det\mathbf{B}_i^k - \log\det\mathbf{B}| \leq \varepsilon$.

---

## D The computing time and the explanation of one-step dual ascent

**Computing time** As our algorithm is based on Bayesian learning and involves covariance structures estimation, it is slower compared to heuristic algorithm StructOMP and solvers like CVX and SPGL1

(group Lasso, group BPDN). Here, we provide curves of NMSE versus CPU time for DivSBL, DivSBL (with complete dual ascent), and BSBL to better visualize the speed of the algorithms. In Figure 9, DivSBL shows larger NMSE reductions at each time step compared to both BSBL and fully iterative DivSBL.

**One-step dual ascent**  It has been observed that imposing strong constraints $\mathbf{B}_i = \mathbf{B}$ leads to slow convergence, while not applying any constraints results in overfitting [24].

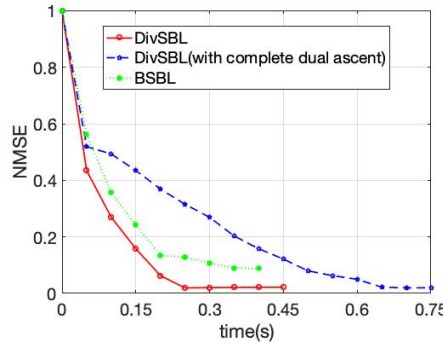

Therefore, by establishing weak constraints and employing dual ascent to solve the constrained optimization problem (P), we achieve diversification on intra-block correlation matrices, which are more aligned with real-life modeling. Since the convergence speed of dual ascent is fastest in the initial few steps and sub-problems are unnecessary to be solved accurately, our initial motivation was to consider allowing it to iterate only once, which yielded promising experimental results, as shown in the Figure 9.

Figure 9: Comparison of Computation Time

We consider providing further theoretical and intuitive explanations. From another perspective, the tuple $(\mathbf{B}_i^{k+1}, \lambda_i^k)$ satisfying the update formula (23)(24) in the text is, in fact, a KKT pair of a certain constrained optimization problem, which is summarized in Proposition 3.1.

**Proof of Proposition 3.1**

*Proof.* Construct a weak constraint function $\psi : \mathbb{R}^{n^2} \to \mathbb{R}$ such that:

$$\nabla \psi(\mathbf{B}_i^{k+1}) = \nabla \zeta(\mathbf{B}_i^{k+1})$$
$$\psi(\mathbf{B}_i^{k+1}) = \psi(\mathbf{B})$$

Such a function $\psi$ always exists. In fact, we can always construct a polynomial function $\psi$ that satisfies the given conditions (Hermitte interpolation polynomial). The finite conditions of function values and derivatives correspond to a system of linear equations for the coefficients of the polynomial function. Since the degree of the polynomial function is arbitrary, we can always ensure that the system of linear equations has a solution by increasing the degree.

Since $\nabla_{\mathbf{B}_i} Q(\{\mathbf{B}_i^{k+1}\}_{i=1}^g, \{\mathbf{G}_i\}_{i=1}^g) - \lambda_i^k \nabla \zeta(\mathbf{B}_i^{k+1}) = 0$, we have:

$$\nabla_{\mathbf{B}_i} Q(\{\mathbf{B}_i^{k+1}\}_{i=1}^g, \{\mathbf{G}_i\}_{i=1}^g) - \lambda_i^k \nabla \psi(\mathbf{B}_i^{k+1}) = 0$$
$$\psi(\mathbf{B}_i^{k+1}) = \psi(\mathbf{B})$$

Thus, $(\{\mathbf{B}_i^{k+1}\}_{i=1}^g, \{\lambda_i^k\}_{i=1}^g)$ forms a KKT pair of the above optimization problem. □

From the proposition above, it can be observed that the tuple $(\mathbf{B}_i^{k+1}, \lambda_i^k)$ satisfying equation (23)(24) is, in fact, a KKT point of a certain weakly constrained optimization problem. Therefore, although we do not precisely solve the constrained optimization problem under the weak constraint $\zeta(\cdot)$ (i.e., $\log \det(\cdot)$ in the main text), we accurately solve the constrained optimization problem under the weak constraint $\psi(\cdot)$. Even though this weak constraint $\psi(\cdot)$ is unknown, it certainly exists and may be better than the original weak constraint $\zeta(\cdot)$. This perspective aligns with the concept of inverse optimization [35]. Interestingly, the technique here provides an application case for inverse optimization. From this viewpoint, we are still optimizing $Q$ function with the weak constraint $\psi(\cdot)$, and the subproblems under the weak constraint $\psi(\cdot)$ are accurately solved. Therefore, similar to the constrained EM algorithm, it can still generate a sequence of solutions, which explains the experimental results mentioned above.

Meanwhile, the above method can also be viewed as a regularized EM algorithm [36]. The update formula (23) corresponds to the exact solution of the regularized $Q$ function $Q(\{\mathbf{B}_i\}_{i=1}^g, \{\mathbf{G}_i\}_{i=1}^g) -$

$\sum_i \lambda_i^k(\zeta(\mathbf{B}_i) - \zeta(\mathbf{B}))$, while equation (24) utilizes gradient descent to update the regularization parameters.

## E  Proof of Theorem 4.1

*Proof.* The posterior estimation of the block sparse signal is given by $\hat{\mathbf{x}} = \beta\hat{\mathbf{\Sigma}}\mathbf{\Phi}^T\mathbf{y} = \left(\beta^{-1}\hat{\mathbf{\Sigma}}_0^{-1} + \mathbf{\Phi}^T\mathbf{\Phi}\right)^{-1}\mathbf{\Phi}^T\mathbf{y}$, where $\hat{\mathbf{\Sigma}}_0 = \text{diag}\left\{\hat{\mathbf{G}}_1\hat{\mathbf{B}}_1\hat{\mathbf{G}}_1, \hat{\mathbf{G}}_2\hat{\mathbf{B}}_2\hat{\mathbf{G}}_2, \cdots, \hat{\mathbf{G}}_g\hat{\mathbf{B}}_g\hat{\mathbf{G}}_g\right\}$, and $\hat{\mathbf{G}}_i = \text{diag}\{\sqrt{\hat{\gamma}_{i1}}, \cdots, \sqrt{\hat{\gamma}_{iL}}\}$.

Let $\hat{\boldsymbol{\gamma}} = (\hat{\gamma}_{11}, \cdots, \hat{\gamma}_{1L}, \cdots, \hat{\gamma}_{g1}, \cdots, \hat{\gamma}_{gL})^T$, which is obtained by globally minimizing (35) for a given $\hat{\mathbf{B}}_i$ ($\forall i$).

$$\min_{\boldsymbol{\gamma}} \mathcal{L}(\boldsymbol{\gamma}) = \mathbf{y}^T\mathbf{\Sigma}_y^{-1}\mathbf{y} + \log\det\mathbf{\Sigma}_y. \tag{35}$$

Inspired by [37], we can rewrite the first summation term as

$$\mathbf{y}^T\mathbf{\Sigma}_y^{-1}\mathbf{y} = \min_{\mathbf{x}}\left\{\beta\|\mathbf{y} - \mathbf{\Phi}\mathbf{x}\|_2^2 + \mathbf{x}^T\mathbf{\Sigma}_0^{-1}\mathbf{x}\right\}.$$

Then (35) is equivalent to

$$\min_{\boldsymbol{\gamma}} \mathcal{L}(\boldsymbol{\gamma}) = \min_{\boldsymbol{\gamma}}\left\{\min_{\mathbf{x}}\left\{\beta\|\mathbf{y} - \mathbf{\Phi}\mathbf{x}\|_2^2 + \mathbf{x}^T\mathbf{\Sigma}_0^{-1}\mathbf{x}\right\} + \log\det\mathbf{\Sigma}_y\right\}$$

$$= \min_{\mathbf{x}}\left\{\min_{\boldsymbol{\gamma}}\left\{\mathbf{x}^T\mathbf{\Sigma}_0^{-1}\mathbf{x} + \log\det\mathbf{\Sigma}_y\right\} + \beta\|\mathbf{y} - \mathbf{\Phi}\mathbf{x}\|_2^2\right\}.$$

So when $\beta \to \infty$, (35) is equivalent to minimizing the following problem,

$$\min_{\mathbf{x}}\left\{\min_{\boldsymbol{\gamma}}\left\{\mathbf{x}^T\mathbf{\Sigma}_0^{-1}\mathbf{x} + \log\det\mathbf{\Sigma}_y\right\}\right\} \tag{36}$$
$$\text{s.t.} \quad \mathbf{y} = \mathbf{\Phi}\mathbf{x}.$$

Let $g(\mathbf{x}) = \min_{\boldsymbol{\gamma}}\left(\mathbf{x}^T\mathbf{\Sigma}_0^{-1}\mathbf{x} + \log\det\mathbf{\Sigma}_y\right)$, then according to Lemma 1 in [37], $g(\mathbf{x})$ satisfies

$$g(\mathbf{x}) = \mathcal{O}(1) + [M - \min(M, KL)]\log\beta^{-1},$$

where $K$ represents the estimated number of blocks and $\beta^{-1} \to 0$ (noiseless). Therefore, when $g(\mathbf{x})$ achieves its minimum value by (36), $K$ will achieve its minimum value simultaneously.

The results in [38] demonstrate that if $K_0 < \frac{M+1}{2L}$, then no other solution exists such that $\mathbf{y} = \mathbf{\Phi}\mathbf{x}$ with $K < \frac{M+1}{2L}$. Therefore, we have $K \geq K_0$, and when $K$ reaches its minimum value $K_0$, the estimated signal $\hat{\mathbf{x}} = \mathbf{x}_{\text{true}}$. $\qquad\square$

## F  Proof of Lemma 4.2

*Proof.* We can equivalently transform the constraint $\mathbf{Z} \succeq \mathbf{\Phi}\mathbf{\Sigma}_0\mathbf{\Phi}^T + \beta^{-1}\mathbf{I}$ into

$$\mathbf{Z} \succeq \mathbf{\Phi}\mathbf{\Sigma}_0\mathbf{\Phi}^T + \beta^{-1}\mathbf{I} \Longleftrightarrow \forall\boldsymbol{\omega} \in \mathbb{R}^m, \quad \boldsymbol{\omega}^T\mathbf{Z}\boldsymbol{\omega} \geq \boldsymbol{\omega}^T\mathbf{\Phi}\mathbf{\Sigma}_0\mathbf{\Phi}^T\boldsymbol{\omega} + \beta^{-1}\boldsymbol{\omega}^T\boldsymbol{\omega}.$$

The LHS $\boldsymbol{\omega}^T\mathbf{Z}\boldsymbol{\omega}$ is linear with respect to $\mathbf{Z}$. And for the RHS, $\boldsymbol{\omega}^T\mathbf{\Phi}\mathbf{\Sigma}_0\mathbf{\Phi}^T\boldsymbol{\omega} + \beta^{-1}\boldsymbol{\omega}^T\boldsymbol{\omega} = \mathbf{q}^T\mathbf{\Sigma}_0\mathbf{q} + \beta^{-1}\boldsymbol{\omega}^T\boldsymbol{\omega}$, where $\mathbf{q} = \mathbf{\Phi}^T\boldsymbol{\omega}$, and $\mathbf{\Sigma}_0$ can bu reformulated as

$$\mathbf{\Sigma}_0 = \text{diag}(\sqrt{\gamma_{11}}\dots\sqrt{\gamma_{gL}})\tilde{\mathbf{B}}\,\text{diag}(\sqrt{\gamma_{11}}\dots\sqrt{\gamma_{gL}})$$

$$= \begin{pmatrix} \gamma_{11}\tilde{B}_{11} & \sqrt{\gamma_{11}}\sqrt{\gamma_{12}}\tilde{B}_{12} & \cdots & \sqrt{\gamma_{11}}\sqrt{\gamma_{gL}}\tilde{B}_{1N} \\ \vdots & \vdots & & \vdots \\ \sqrt{\gamma_{gL}}\sqrt{\gamma_{11}}\tilde{B}_{N1} & \sqrt{\gamma_{gL}}\sqrt{\gamma_{gL}}\tilde{B}_{N2} & \cdots & \gamma_{gL}\tilde{B}_{NN} \end{pmatrix}.$$

Therefore, RHS$= q_1^2\tilde{B}_{11}\gamma_{11} + \dots + q_1q_N\tilde{B}_{1N}\sqrt{\gamma_{gL}}\sqrt{\gamma_{11}} + \dots + q_Nq_1\tilde{B}_{N1}\sqrt{\gamma_{11}}\sqrt{\gamma_{gL}} + q_N^2\tilde{B}_{NN}\gamma_{gL} + \beta^{-1}\boldsymbol{\omega}^T\boldsymbol{\omega} = \text{vec}(\mathbf{q}\mathbf{q}^T \odot \tilde{\mathbf{B}})^T(\sqrt{\boldsymbol{\gamma}} \otimes \sqrt{\boldsymbol{\gamma}}) + \beta^{-1}\boldsymbol{\omega}^T\boldsymbol{\omega}$ which is linear with respect to $\sqrt{\boldsymbol{\gamma}} \otimes \sqrt{\boldsymbol{\gamma}}$. In conclusion, $\mathbf{Z} \succeq \mathbf{\Phi}\mathbf{\Sigma}_0\mathbf{\Phi}^T + \beta^{-1}\mathbf{I}$ is convex with respect to $\mathbf{Z}$ and $\sqrt{\boldsymbol{\gamma}} \otimes \sqrt{\boldsymbol{\gamma}}$,

$\qquad\square$

# G Proof of Lemma 4.3

*Proof.* " $\implies$ " Given $\mathbf{y}^T \boldsymbol{\Sigma}_y^{-1} \mathbf{y} = C$ and $\mathbf{u}$ satisfying $\mathbf{y}^T \mathbf{u} = C$, without loss of generality, we choose $\mathbf{u} \triangleq \boldsymbol{\Sigma}_y^{-1} \mathbf{y}$, i.e., $\mathbf{y} = \boldsymbol{\Sigma}_y \mathbf{u}$. Then $\mathbf{b}$ can be rewritten as

$$\mathbf{b} = \left( \boldsymbol{\Sigma}_y - \beta^{-1} \mathbf{I} \right) \mathbf{u} = \boldsymbol{\Phi} \boldsymbol{\Sigma}_0 \boldsymbol{\Phi}^T \mathbf{u}. \tag{37}$$

Applying the vectorization operation to both sides of the equation, (37) results in

$$\mathbf{b} = \mathrm{vec} \left( \boldsymbol{\Phi} \boldsymbol{\Sigma}_0 \boldsymbol{\Phi}^T \mathbf{u} \right) = \left[ (\mathbf{u}^T \boldsymbol{\Phi}) \otimes \boldsymbol{\Phi} \right] \mathrm{vec} \left( \boldsymbol{\Sigma}_0 \right)$$

$$= \left[ (\mathbf{u}^T \boldsymbol{\Phi}) \otimes \boldsymbol{\Phi} \right] \mathrm{vec} \left( \tilde{\mathbf{G}} \tilde{\mathbf{B}} \tilde{\mathbf{G}} \right)$$

$$= \left[ (\mathbf{u}^T \boldsymbol{\Phi}) \otimes \boldsymbol{\Phi} \right] \left( \tilde{\mathbf{G}} \otimes \tilde{\mathbf{G}} \right) \mathrm{vec} \left( \tilde{\mathbf{B}} \right)$$

$$= \left[ (\mathbf{u}^T \boldsymbol{\Phi}) \otimes \boldsymbol{\Phi} \right] \mathrm{diag} \left( \mathrm{vec} \left( \tilde{\mathbf{B}} \right) \right) \mathrm{Diag} \left( \tilde{\mathbf{G}} \otimes \tilde{\mathbf{G}} \right)$$

$$= \left[ (\mathbf{u}^T \boldsymbol{\Phi}) \otimes \boldsymbol{\Phi} \right] \mathrm{diag} \left( \mathrm{vec} \left( \tilde{\mathbf{B}} \right) \right) \cdot \left( \sqrt{\boldsymbol{\gamma}} \otimes \sqrt{\boldsymbol{\gamma}} \right).$$

" $\impliedby$ " Vice versa. $\qquad \square$

# H Proof of Theorem 4.4

*Proof.* Based on Lemma 4.3, we consider the following optimization problem:

$$\min_{\boldsymbol{\gamma}} \quad \log \det \boldsymbol{\Sigma}_y$$
$$\text{s.t.} \quad \mathbf{P} \left( \sqrt{\boldsymbol{\gamma}} \otimes \sqrt{\boldsymbol{\gamma}} \right) = \mathbf{b} \tag{38}$$
$$\boldsymbol{\gamma} \succeq \mathbf{0},$$

where $\boldsymbol{\Sigma}_y = \beta^{-1} \mathbf{I} + \boldsymbol{\Phi} \boldsymbol{\Sigma}_0 \boldsymbol{\Phi}^T$, $\mathbf{P}$ and $\mathbf{b}$ are already defined in Lemma 4.3. In order to analyze the property of the minimization problem (38), we introduce a symmetric matrix $\mathbf{Z} \in \mathbb{R}^{M \times M}$ here. Therefore, the problem with respect to $\mathbf{Z}$ and $\boldsymbol{\gamma}$ becomes

$$\min_{\mathbf{Z}, \boldsymbol{\gamma}} \quad \log \det \mathbf{Z} \tag{a.1}$$
$$\text{s.t.} \quad \mathbf{Z} \succeq \boldsymbol{\Phi} \boldsymbol{\Sigma}_0 \boldsymbol{\Phi}^T + \beta^{-1} \mathbf{I} \tag{a.2}$$
$$\mathbf{P} \left( \sqrt{\boldsymbol{\gamma}} \otimes \sqrt{\boldsymbol{\gamma}} \right) = \mathbf{b} \tag{a.3}$$
$$\boldsymbol{\gamma} \succeq \mathbf{0}, \tag{a.4}$$

It is evident that problem (38) and problem (a) are equivalent. Denote the solution of (a) as $(\mathbf{Z}^*, \boldsymbol{\gamma}^*)$, so $\boldsymbol{\gamma}^*$ here is also the solution of (38). Thus, we will analysis the minimization problem (a) instead in the following paragraph.

We first demonstrate the concavity of (a). Obviously, with respect to $\mathbf{Z}$ and $\left( \sqrt{\boldsymbol{\gamma}} \otimes \sqrt{\boldsymbol{\gamma}} \right)$, the objective function $\log \det \mathbf{Z}$ is concave, and (a.3) is convex. According to Lemma 4.2, (a.2) is convex as well. Hence, we only need to show the convexity of (a.4) with respect to $\mathbf{Z}$ and $\left( \sqrt{\boldsymbol{\gamma}} \otimes \sqrt{\boldsymbol{\gamma}} \right)$.

It is observed that

$$\boldsymbol{\gamma} = \begin{pmatrix} \mathbf{h}_1^T & & & \\ & \mathbf{h}_2^T & & \\ & & \ddots & \\ & & & \mathbf{h}_{gL}^T \end{pmatrix} \left( \sqrt{\boldsymbol{\gamma}} \otimes \sqrt{\boldsymbol{\gamma}} \right) \triangleq \mathbf{H} \left( \sqrt{\boldsymbol{\gamma}} \otimes \sqrt{\boldsymbol{\gamma}} \right),$$

where $\mathbf{h}_i \triangleq (\delta_{i1}, \cdots, \delta_{i,gL})^T$, $\mathbf{H} \in \mathbb{R}^{gL \times (gL)^2}$. Based on the convexity-preserving property of linear transformation [39], the constraint $\boldsymbol{\gamma} \succeq \mathbf{0}$ exhibits convexity with respect to $\left( \sqrt{\boldsymbol{\gamma}} \otimes \sqrt{\boldsymbol{\gamma}} \right)$. Therefore, (a) is concave with respect to $\left( \sqrt{\boldsymbol{\gamma}} \otimes \sqrt{\boldsymbol{\gamma}} \right)$ and $\mathbf{Z}$. So we can rewrite (a) as

$$\min_{\mathbf{Z}, \sqrt{\boldsymbol{\gamma}} \otimes \sqrt{\boldsymbol{\gamma}}} \quad \log \det \mathbf{Z}$$
$$\text{s.t.} \quad \mathbf{Z} \succeq \boldsymbol{\Phi} \boldsymbol{\Sigma}_0 \boldsymbol{\Phi}^T + \beta^{-1} \mathbf{I}$$
$$\mathbf{P} \left( \sqrt{\boldsymbol{\gamma}} \otimes \sqrt{\boldsymbol{\gamma}} \right) = \mathbf{b} \tag{b}$$
$$\boldsymbol{\gamma} \succeq \mathbf{0}.$$

The minimum of (b) will achieve at an extreme point. According to the equivalence between extreme point and basic feasible solution (BFS) [40], the extreme point of (b) is a BFS to

$$\begin{cases} \mathbf{Z} \succeq \mathbf{\Phi}\mathbf{\Sigma}_0\mathbf{\Phi}^T + \beta^{-1}\mathbf{I} \\ \mathbf{P}\left(\sqrt{\gamma} \otimes \sqrt{\gamma}\right) = \mathbf{b} \\ \gamma \succeq \mathbf{0} \end{cases},$$

which concludes $||\sqrt{\gamma} \otimes \sqrt{\gamma}||_0 \leq r(\mathbf{P}) = M$, equivalently $||\gamma||_0 \leq \sqrt{M}$. This result implies that every local minimum (also a BFS to the convex polytope) must be attained at a sparse solution. □

# I The experiment of 1D signals with block sparsity

## I.1 The reconstruction results

As mentioned in Section 2.1.3, homoscedasticity can be seen as a special case of our model. Therefore, we test our algorithm on homoscedastic data provided in [24]. In this dataset, each block shares the same size $L = 6$, and the amplitudes within each block follow a homoscedastic normal distribution.

The reconstructed results are shown in Figure 10, Figure 3 and the first part of Table 1. DivSBL demonstrates a significant improvement compared to other algorithms.

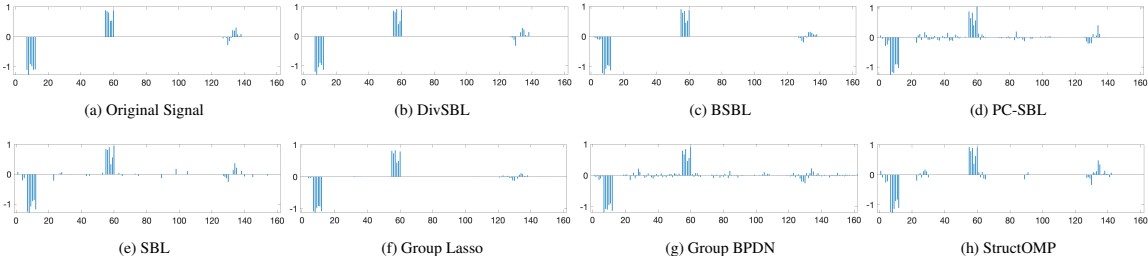

Figure 10: The original **homoscedastic** signal and reconstructed results by various algorithms. (N=162, M=80)

Furthermore, we consider heteroscedastic signal, which better reflects the characteristics of real-world data. The recovery results are presented in Figure 11, Figure 3 and the second part of Table 1.

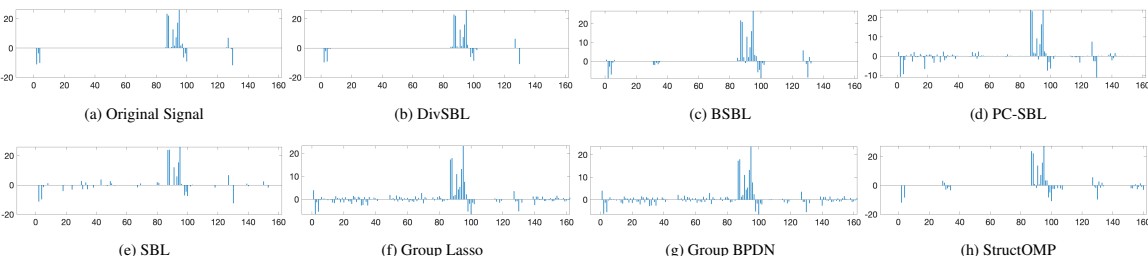

Figure 11: The original **heteroscedastic** signal and reconstructed results by various algorithms.(N=162, M=80)

## I.2 Reconstructed by Bayesian methods with credible intervals for point estimation

Here we provide comparative experiments on heteroscedastic data with three classic Bayesian sparse regression methods: Horseshoe model [41], spike-and-slab Lasso [42] and hierarchical normal-gamma method [43] in Figure 12. Regarding the natural advantage of Bayesian methods in quantifying uncertainties for point estimates, we further include the posterior confidence intervals from Bayesian methods. As shown in Figure 13, DivSBL offers more stable and accurate posterior confidence intervals.

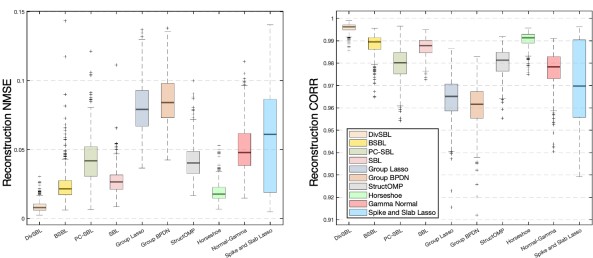

Figure 12: Reconstruction error (NMSE) and correlation.

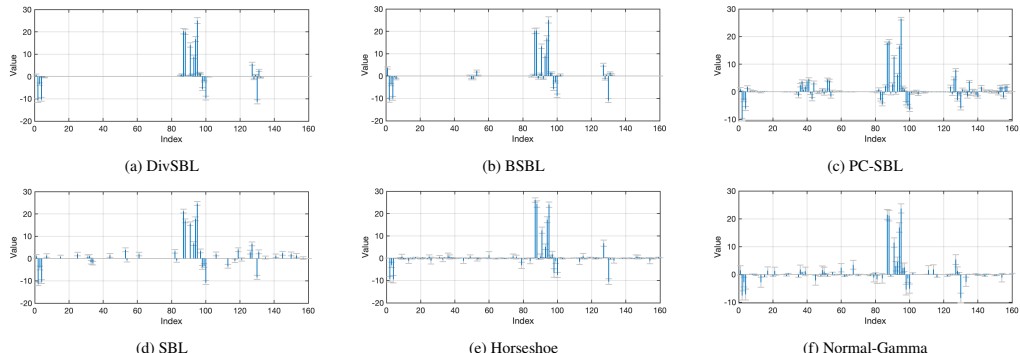

Figure 13: Confidence intervals for each Bayesian approach.

## J  The experiment of audio signals

Audio signals display block sparse structures in the discrete cosine transform (DCT) basis. As illustrated in Figure 14, the original audio signal (a) transforms into a block sparse structure (b) after DCT transformation.

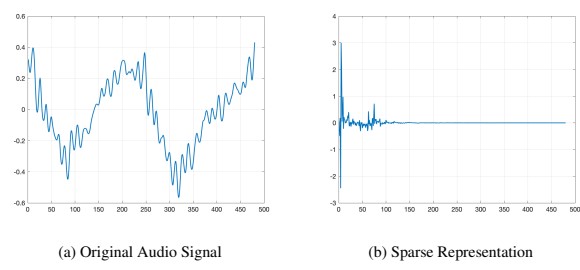

(a) Original Audio Signal      (b) Sparse Representation

Figure 14: The original signal and its sparse representation.

We carry out experiments on real-world audio signal, which is randomly chosen in *AudioSet* [34]. The reconstruction results for audio signals are present in Figure 15. It is noteworthy that DivSBL exhibits an improvement of **over 24.2%** compared to other algorithms.

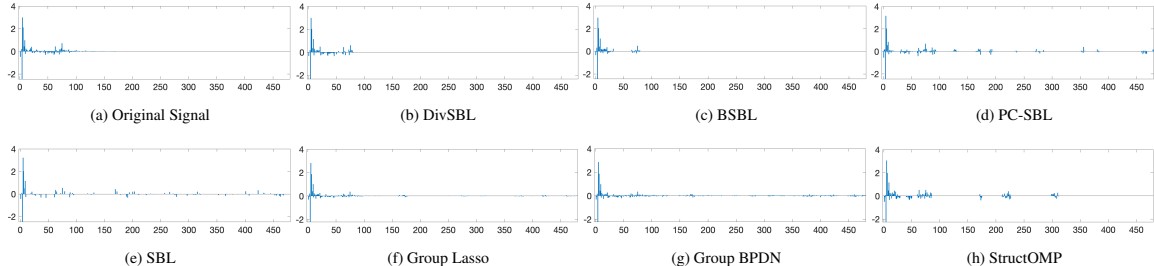

(a) Original Signal    (b) DivSBL    (c) BSBL    (d) PC-SBL

(e) SBL    (f) Group Lasso    (g) Group BPDN    (h) StructOMP

Figure 15: The sparse audio signal reconstructed by respective algorithms. NMSE: (b) **0.0406** (c) 0.0572 (d) 0.1033 (e) 0.1004 (f) 0.0536 (g) 0.0669 (h) 0.1062. ($N = 480, M = 150$)

**The sensitivity of sample rate**  As demonstrate in Section 5.3, we tested on audio sets to investigate the sensitivity of sample rate ($M/N$) varied from 0.25 to 0.55. DivSBL emerges as the top performer across diverse sampling rates, exhibiting a consistent 1 dB improvement in NMSE relative to the best-performing algorithm, as depicted in Figure 16.

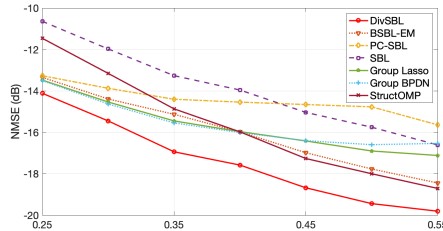

Figure 16: NMSE vs. sample rates.

## K  The experiment of image reconstruction

As depicted in Figure 17, the images exhibit block sparsity in discrete wavelet domain. We've created box plots for NMSE and correlation reconstruction results for each image undergoing restoration,

as depicted in Figures 18–25. It's evident that the DivSBL exhibits significant advantages in image reconstruction tasks.

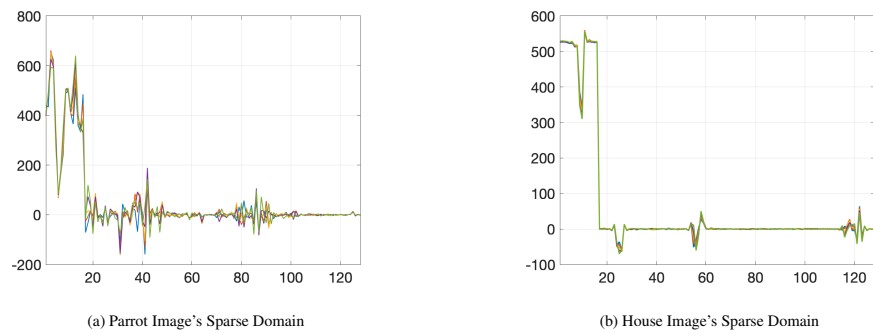

(a) Parrot Image's Sparse Domain

(b) House Image's Sparse Domain

Figure 17: Parrot and House image data (the first five columns) transformed in discrete wavelet domain.

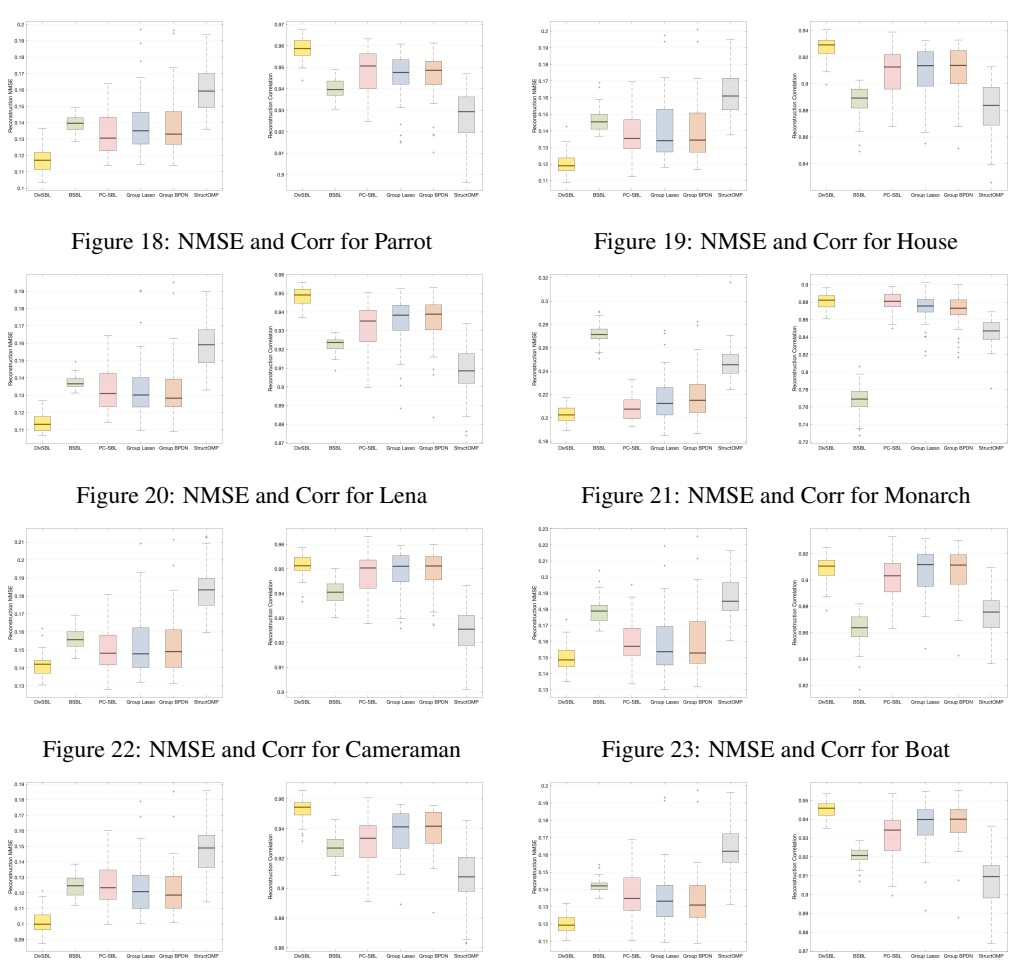

Figure 18: NMSE and Corr for Parrot

Figure 19: NMSE and Corr for House

Figure 20: NMSE and Corr for Lena

Figure 21: NMSE and Corr for Monarch

Figure 22: NMSE and Corr for Cameraman

Figure 23: NMSE and Corr for Boat

Figure 24: NMSE and Corr for Foreman

Figure 25: NMSE and Corr for Barbara

## L    The sensitivity to initialization

According to our algorithm, given the variance $\gamma_{ij}$, the prior covariance matrix can be obtained as $\mathbf{\Sigma}_0 = \mathrm{diag}(\sqrt{\gamma_{11}}, \cdots, \sqrt{\gamma_{gL}})\tilde{\mathbf{B}}\mathrm{diag}(\sqrt{\gamma_{11}}, \cdots, \sqrt{\gamma_{gL}})$. In the absence of any structural infor-

mation, the initial correlation matrix $\tilde{\mathbf{B}}$ is set to the identity matrix. Consequently, the mean and covariance matrix for the first iteration can also be determined. Since other variables are derived from the variance, we only need to test the sensitivity to the initial values of variances $\boldsymbol{\gamma}$.

We test the sensitivity of DivSBL to initialization on the heteroscedastic signal from Section 5.1. Initial variances are set to $\boldsymbol{\gamma} = \eta \cdot \text{ones}(gL, 1)$ and $\boldsymbol{\gamma} = \eta \cdot \text{rand}(gL, 1)$ with the scale parameter $\eta$ ranging from $1 \times 10^{-1}$ to $1 \times 10^{4}$. The result in Figure 26 shows that while initialization could affect the convergence speed to some extent, the algorithm's overall convergence is assured.

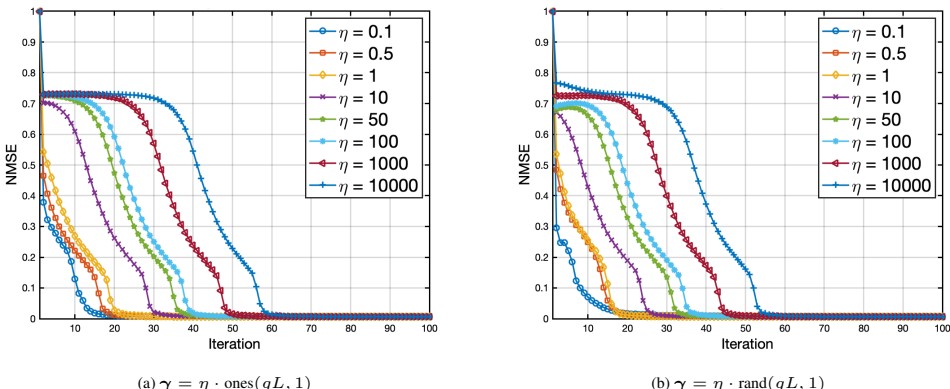

(a) $\boldsymbol{\gamma} = \eta \cdot \text{ones}(gL, 1)$

(b) $\boldsymbol{\gamma} = \eta \cdot \text{rand}(gL, 1)$

Figure 26: The sensitivity to initialization for DivSBL.

