# OpenReview forum: "Block Sparse Bayesian Learning: A Diversified Scheme"
_NeurIPS.cc/2024/Conference — NeurIPS 2024 poster_

### Official Review · Reviewer_KPTu · 2024-07-07

**Soundness:** 3
**Presentation:** 3
**Contribution:** 2
**Rating:** 6
**Confidence:** 3

**Summary:**

The paper introduces a prior named Diversified Block Sparse Prior, which can be viewed as a generalization of priors for existing sparse Bayesian learning methods. It utilizes EM algorithm and dual ascent to obtain parameter estimates and convergence of estimates to the true parameter in the $\beta$ limit has been established.

**Strengths:**

The structure of the paper is well organized for readers to follow. Generalization of the existing sparse Bayesian learning methods would provide more flexible framework in sparse regression problems.

**Weaknesses:**

1. The paper does not mention any of the existing well-known Bayesian sparse regression methods: for instance, horse-shoe prior [1], spike-slab LASSO [2], hierarchical normal-gamma hyperpriors [3], which have been very successful in sparse signal estimation problems. Furthermore, the strength of the Bayesian approach in this problem is in quantifying uncertainties associated with the point estimate. Unlike many frequentist approaches where the construction of confidence intervals requires a more sophisticated debiasing approach, the Bayesian approach provides a natural way to obtain credible intervals for the point estimate. The paper seems to largely ignore such strength of the Bayesian approach.

[1] Carvalho, Carlos M., Nicholas G. Polson, and James G. Scott. "Handling sparsity via the horseshoe." Artificial intelligence and statistics. PMLR, 2009.

[2] Ročková, Veronika, and Edward I. George. "The spike-and-slab lasso." Journal of the American Statistical Association 113.521 (2018): 431-444

[3] Calvetti, Daniela, Erkki Somersalo, and A. Strang. "Hierachical Bayesian models and sparsity: ℓ2-magic." Inverse Problems 35.3 (2019): 035003.

2. The paper establishes theoretical results on local minima in Section 4.2, but this section doesn't seem to add much information about the algorithm. Perhaps the authors should provide more qualitative statements from the established theoretical results, which the current manuscript misses.

3. The current theoretical guarantee only considers the noiseless setting. Is there anything further one could say in the noisy setting?

**Questions:**

See above.

**Limitations:**

Not Applicable.

---

> ### Author Rebuttal · Authors · 2024-08-07
>
> We sincerely appreciate your feedback and constructive suggestions on our paper, which I believe will help to enrich the content of the original text. In this rebuttal, we respond to the concerns raised in the reviews.
>
> ---
>
> ### **Q1**:
> Thank you for your suggestions. Since our paper primarily investigates structured block sparsity, we did not pay much attention to purely sparse models. Horseshoe model, spike and slab LASSO, and hierarchical normal-gamma model are indeed classic Bayesian sparse regression methods. Given that our paper is also based on the Bayesian framework, we deeply agree that these classic works should be mentioned, and we are willing to cite these seminal works in the revised version of our paper.
>
> Additionally, **in the global rebuttal PDF**, we have included comparative experiments (Section 5.1 of the paper) with these three classic methods **in Fig.4**, further illustrating the advantages of DivSBL in block sparse recovery problems. Regarding your point about the natural advantage of Bayesian methods in quantifying uncertainties, we have included the posterior confidence intervals of Bayesian methods **in Fig.3 of the global rebuttal PDF**. As shown in the figure, DivSBL provides more stable and accurate posterior confidence intervals. We will incorporate this point into the revised paper. Once again, we appreciate your constructive feedback.
>
> ---
>
> ### **Q2**:
> This is a very good question. The theory established in Section 4 is primarily intended to benchmark against BSBL. As a classic work in the field, BSBL has both theoretical and experimental guarantees. In Section 4, we demonstrate that although the DivSBL model is more complex compared to BSBL, with more latent variables and higher nonlinearity, it still has similar or even better theoretical properties than the BSBL model. (Another classic work in the field, PC-SBL, does not have such theoretical properties.) However, at the algorithmic level, for both BSBL and DivSBL, the conditions under which the algorithm converges to global or local optima have not been established. This is because in non-convex optimization models, whether and when the EM algorithm converges to a global or local solution has been a long-standing problem in the optimization field, although in our experiments, we generally obtain satisfactory solutions in most cases, which, according to Theorem 4.1, are typically global minima. Fortunately, there is now some work that indicates for most common non-convex objective functions in machine learning, most local minima are approximately global minima [1]. However, this is well beyond the scope of this paper.
>
> ---
>
> ### **Q3**:
> We sincerely thank the reviewer for your insightful question, which has driven us to further our theoretical advancements. For the global minimum, the proof is a generalization of [2][3], and it seems hard to relax the noiseless condition due to the fact that the equivalent transformations from (35) to (36) in the paper depend on the assumption of noiselessness.
>
> For the local minimum, we initially used the Schur complement to prove Lemma 4.2, which led to the proof of Theorem 4.4, but this proof was only applicable to the noiseless scenario. We subsequently adjusted the proof, with the core being the direct proof of lines 493-494 in Appendix H of the original paper without relying on Lemma 4.2, allowing the method to be extended to noisy scenarios. This also validates our conjecture in the original paper. **For a detailed proof, please refer to the global rebuttal**.
>
> ---
>
> References:
>
> [1] Ma T. Why do local methods solve nonconvex problems? [J]. 2020.
>
> [2] Wipf D P, Rao B D. Sparse Bayesian learning for basis selection[J]. IEEE Transactions on Signal processing, 2004, 52(8): 2153-2164.
>
> [3] Zhang Z, Rao B D. Sparse signal recovery with temporally correlated source vectors using sparse Bayesian learning[J]. IEEE Journal of Selected Topics in Signal Processing, 2011, 5(5): 912-926.

---

> > ### Comment · Reviewer_KPTu · 2024-08-12
> > **Response**
> >
> > Thanks for the detailed response in addressing several questions I had. I have adjusted the score based on the detailed response.

---

> > > ### Author Response · Authors · 2024-08-12
> > > **Thank you**
> > >
> > > Thank you for your recognition! We will incorporate the content of this rebuttal in the latest version of the paper. Once again, we appreciate your valuable feedback!

---

### Official Review · Reviewer_c5nz · 2024-07-12

**Soundness:** 4
**Presentation:** 4
**Contribution:** 3
**Rating:** 7
**Confidence:** 3

**Summary:**

This paper introduces block sparse bayesian learning for block sparse settings, as motivated by compressed sensing theory. The method relies on a “diversified scheme” which allows for inference that is robust to block choices by modeling intra-block covariance and inter-block correlation. The authors derive an algorithm for learning the model parameters and provide some simple but solid convergence results. They demonstrate their method’s effectiveness on a variety of applications, showing it is more robust than multiple existing and established methods to the prior specification of block information.

**Strengths:**

The method introduced is intuitive. The authors provide a clear description of the EM algorithm and dual ascent method, backing the use of both with theoretical justification. Impressive ground truth and computational results, a technically sound paper overall.

**Weaknesses:**

Since EM is used for fitting, method may be sensitive to initialization.

**Questions:**

In practice, when the block structure is misspecified, how well does the proposed method learn the zero variance terms described in e.g. figure 2? How sensitive is the method to initialization?

**Limitations:**

The authors evaluate the limitations of their work with extensive experiments, as outlined in the checklist. However, the authors could do a better job of addressing the limitations of their work (for example, potential sensitivity to initialization) directly within their paper.

---

> ### Author Rebuttal · Authors · 2024-08-07
>
> We deeply appreciate your kind words regarding the clarity of our presentation and the recognition of our theoretical and experimental work. Additionally, we value your insightful questions about initialization and the practical demonstration of the variance term. We will incorporate the content of this rebuttal in the subsequent version of the paper.
>
> ---
>
> ### **Q1 (Sensitivity to initialization):**
> Thank you for your thorough understanding of DivSBL and your insightful questions. Following your suggestion, we tested the impact of initialization on the DivSBL algorithm. According to our algorithm, given the variance $\{\gamma_{ij}\}$, the prior covariance matrix can be obtained as
> $\Sigma_0 = \text{diag}(\sqrt{\gamma_{11}}, \cdots, \sqrt{\gamma_{gL}}) B \text{diag}(\sqrt{\gamma_{11}}, \cdots, \sqrt{\gamma_{gL}})$.
> In the absence of any structural information, the initial correlation matrix $B$ is set to the identity matrix. Consequently, the mean and covariance matrix for the first iteration can also be determined. Since other variables are derived from the variance, we only need to test the sensitivity to the initial values of variances $\gamma$.
>
> **The results are displayed in Fig.5 in the global rebuttal PDF**. Fig. 5(a) shows the iteration curve with the initial variance vector
> $
> \gamma = \eta \cdot \text{ones}(g L, 1)
> $
> while Fig. 5(b) shows it with
> $
> \gamma = \eta \cdot \text{rand}(gL, 1)
> $
> The parameter $\eta$ varies from $1 \times 10^{-1}$ to $1 \times 10^{4}$, representing different initial variance values. We observe that while the initialization could affect the convergence rate to some extent, the algorithm's overall convergence is assured.
>
> ---
>
> ### **Q2 (Variance learning):**
> This is a very good question. The variable selection at each position is determined by the shrinkage of the variance at that position. Through variance shrinkage, DivSBL can eventually find the true block structure, making the method robust to the block size.
>
> Based on your constructive questions, we illustrate the structure of the variance learned at different preset block sizes, **as shown in Fig.2 in the global rebuttal PDF**. We find that, regardless of whether the block size is small, medium, or large, DivSBL consistently shrinks to the true block as expected, which is an achievement other block-based algorithms cannot consistently match.

---

> > ### Comment · Reviewer_c5nz · 2024-08-11
> >
> > Thank you to the authors for thoroughly addressing questions and running extensive experiments in the rebuttal stage.

---

> > > ### Author Response · Authors · 2024-08-12
> > > **Thank you**
> > >
> > > Thank you for your recognition! We will incorporate the additional experiments into the latest version of the paper. Once again, we appreciate your valuable feedback!

---

### Official Review · Reviewer_2MRM · 2024-07-21

**Soundness:** 2
**Presentation:** 2
**Contribution:** 2
**Rating:** 4
**Confidence:** 3

**Summary:**

The authors propose a hierarchical bayesian model for sparse inverse problems where sparsity is structured in blocks. The authors propose a diversified block sparse prior using a structured covariance taking into account both intra block and block-to-block correlations. They propose an EM algorithm to solve the problem, showcase some thereotical properties in noiseless settings and illustrate the performance of their model with synthetic and real datasets.

**Strengths:**

- Self contained paper
- Novel and simple model with both theoretical and applied contributions
- The illustration used to provide the intuition are not easy to understand
- The theoretical findings are not well motivated / explained and seem out of place / space fillers.
- The presentation of the paper could be significantly improved (figures hard to read, tiny fonts)

**Weaknesses:**

- The illustration used to provide the intuition are not easy to understand
- The theoretical findings are not well motivated / explained and seem out of place / space fillers.
- The presentation of the paper could be significantly improved (figures hard to read, tiny fonts)

**Questions:**

1. Paragraph 2.1.1 should be explained better. I fail to see how the proposed structured covariance would lead to the diversified sparsity (intra block/extra block) regardless of the predefined blocks. But if the method is robust to the chosen blocks, what is their effect then ? Why not let the method discover the blocks by itself ?

2. I fail to see how the proposed method can recover accurate sparsity structures regardless of the predefined blocks. I do not find fig2 helpful
in this regard. In both conditions (white/pink) the recovered gamma_i are close to 0 ?

3. I believe in  L95 there should be gL*(L+1)/2 constraints ?

4. Is there an intuition behind selecting the constraint function psi ?

5. Where does the toeplitz correction come from ? What is the impact of this step on the convergence of the EM algoritm ? Is this step necessary ?

6. Since the main contribution of the paper is the structured sparsity in groups, in the experiments I would expect the group lasso to perform very well when the groups are known and fixed. Which is not the case here. Is that due to a poor hyperparameter setting ?

- typo: L137 a diversified solution

- typo: L176 the diversified block sparse prior, the following global

---

> ### Author Rebuttal · Authors · 2024-08-07
>
> We sincerely appreciate your feedback and constructive suggestions on our paper, which we believe will help enrich the content of the original text. In this rebuttal, we respond to the concerns raised in the reviews.
>
> ---
>
> ### **Q1**:
> This is a very good question. Block-based methods require a predefined block size and then estimate based on these blocks. Traditional block-based methods are sensitive to predefined block sizes, as they estimate each block as either all zero or all non-zero. If the block size is misspecified, the method can produce significant errors.
>
> In our approach, by diversification, each variable is controlled by a corresponding variance term (while in BSBL, all elements within a block are controlled by a single variance term). Variable selection is based on individual variance shrinkage, avoiding the situation where all elements within a block are simultaneously zero or non-zero, thus making the method robust to the block size.
>
> Although our method still requires a predefined block size, which is necessary because we need to learn a correlation matrix $B_i$ with a determined dimension, the predefined block size acts more like an initial point in DivSBL. Through variance shrinkage, DivSBL can eventually find the true block structure, as you mentioned. You can refer to **Fig. 2 of the global rebuttal PDF**, where regardless of whether we choose a small, medium, or large block size, our method can ultimately identify the true block positions through variance shrinkage, which other block-based methods (like BSBL) do not possess.
>
> ---
>
> ### **Q2**:
> We apologize for the confusion. In Figure 2 of the paper, the ~0 in the pink area indicates that $ \gamma_i $ is non-zero (not close to zero), while the 0 in the white area indicates that $ \gamma_i $ is zero. We want to convey that through diversified variance, we can exclude white areas in predefined blocks via variance shrinkage, thereby adaptively finding the true blocks (i.e., the pink areas), as **shown in Fig. 2 of the global rebuttal PDF**.
>
> In BSBL, all elements within a block share the same $ \gamma $, which means the elements within a block are estimated to be either all zero or all non-zero simultaneously. Interestingly, if we formulate group Lasso in a Bayesian approach, as suggested in [1], it could be found that all elements in the same block also share a common variance in the prior. That’s why traditional block-based methods have this simultaneous zero or non-zero issue. Our insight is that the key to leveraging block information lies in learning the correlation matrix $ B_i$, rather than having the elements within a block share a common variance parameter.
>
> ---
>
> ### **Q3**:
> Yes, although they are of the same order of magnitude, considering the symmetry of the correlation matrix, the number of constraints should be $ \frac{gL(L+1)}{2} $. Thanks for your correction.
>
> ---
>
> ### **Q4**:
> This is a very good question. Reference [2] documents that if correlation matrices $B_i$ of different blocks are not constrained, it could lead to overfitting. Therefore, a strong constraint $ B_i = B $ is used, meaning that each block has the same correlation matrix to avoid overfitting. Figure 7 in the paper shows that the unconstrained algorithm (green line, Diff-BSBL) is faster initially but suffers from significant error increase later due to overfitting. On the other hand, algorithms with strong constraints (black and blue lines, DivSBL without diversified correlation & BSBL) are slower initially but achieve better accuracy later.
>
> Our motivation is to develop an algorithm that achieves both faster speed and better accuracy. Thus, our contribution lies in proposing a weak constraint framework and providing both explicit and implicit formats for choosing $ \psi $, which allows the correlation matrix of each block to retain some similarities while preserving their individual specificities. As shown by the red line (DivSBL) in the figure, DivSBL is faster in the early stages and achieves higher accuracy in the later stages. We believe that there might be better ways to select $ \psi $, and we leave this as future work.
>
> ---
>
> ### **Q5**:
> Thank you for your insightful question. The step of Toeplitz correction originates from BSBL [2] and is necessary, since we need to ensure that $ B_i$ maintains a correlation matrix structure during updates. Therefore, after updating $ B_i $, it needs to be projected onto a correlation matrix. Toeplitz correction provides a feasible and sufficiently simple way to do this, although it does result in some loss of correlation information. We believe that more complex projection methods could be considered, but this is beyond the scope of this paper.
>
> ---
>
> ### **Q6**:
> Thanks for your question. In all of our experiments, the size and location of the blocks are unknown to the algorithm and the classic CVX toolbox is used for solving group Lasso. For block-based methods, we agree that if the blocks are known and fixed, the methods perform well. However, since block information is unknown in real scenarios, block-based methods require preset block information, to which traditional methods are very sensitive, as shown in Figure 4 of the paper.
>
> Our contribution lies in addressing the sensitivity of block-based methods to preset block information via diversification. This way, even if our preset block information is not accurate, we can still adaptively identify the true blocks by variance shrinkage, **as shown in Fig.2 of the global rebuttal PDF**. We further included experiments on the joint effect of noise and the number of observations, demonstrating the advantages of our method in block sparse recovery, **as shown in Fig.1 of the PDF**.
>
> ---
>
>
> References:
>
> [1] Casella G, Ghosh M, Gill J, et al. Penalized regression, standard errors, and Bayesian lassos[J]. 2010.
>
> [2] Zhang Z and Rao B D. Extension of SBL algorithms for the recovery of block sparse signals with intra-block correlation. 2013.

---

### Official Review · Reviewer_rTtn · 2024-07-29

**Soundness:** 3
**Presentation:** 3
**Contribution:** 2
**Rating:** 4
**Confidence:** 4

**Summary:**

In this work, the authors propose a novel prior called Diversified Block Sparse Prior  towards a new framework to address the problem of recovery of block sparse signals. They provide theoretical and experimental justification as a proof of the efficacy of their work.

**Strengths:**

The authors propose a novel diversified block sparse prior which allows for a diversified variance and a diversified correlation. Such a prior can be used to encode/learn the knowledge of DAGs. An EM based solution is derived.

**Weaknesses:**

This area of research is quite old, and hence, any result that comes about tends to have flavors of several previous works. While the results here are relevant for a journal, I do not find anything exciting in the work which is worth publishing at Neurips. Overall the impact of the work is poor. However, in a stand-alone manner, these are some of the weaknesses:
1. The length of the block is assumed constant across the blocks. Although the authors claim that the corresponding entries will be zero or non-zero, the model will be forced towards solutions that have a constant non-zero block size.
2. Lack of sample complexity results: when we introduce new variables into any existing set-up (as compared to B-SBL, the variance here is G_iB_iG_i, which is twice as many parameters) we expect the number of measurements to be larger. Noise also impacts sample complexity. Hence, it is essential to analyse their joint effect on the sample complexity.
3. Local and global minima results hold in no-noise scenarios: The noisy scenario is presented only as a conjecture.
4. Although EM iterations are used, a closed form solution for B_i is not available. Authors propose the ascent approach for estimating B_i. This impacts the estimation process, but it has not been clarified.
5. It is not clear from the experimental settings as to what leads to better results of the proposed algorithm. The algorithm displays better performance in spite of having to estimate additional parameters in G and B. This is possible only if the number of measurements increases. More experimental results based on the ratio, m/p where m denotes the number of observations and p denotes the sparsisity is required.

**Questions:**

1. What is the additional burden due to the introduction of diversified prior. Plots of m/p to substantiate the above.
2. How should the proofs change to include noise?
3. How does your method compare with PC-SBL, to be added into 2.1.3.
4. What is the error in estimating B_i in the synthetic case?

**Limitations:**

This method is effective only with signals that have constant block-size. No sample complexity results are available.

---

> ### Author Rebuttal · Authors · 2024-08-07
>
> Thanks for your valuable questions on our paper. In this rebuttal, we respond to the concerns raised in the reviews.
>
> ---
>
> ### **Q1 (Constant block size)**:
> Thanks for your question. Since the block locations and sizes are unknown in real-world scenarios, block-based methods require a predefined block size to perform estimations. Our observation is that traditional block-based methods are sensitive to the predefined block information because they estimate the elements within a block to be either all zero or all non-zero. As you mentioned, these traditional models will be forced towards solutions that have the constant non-zero predefined block size, leading to significant errors if the block size is misspecified.
>
> In our approach, by diversification, each variable at a position is controlled by a variance term. The variable selection at each position is determined by the shrinkage of the variance at that position. This addresses the simultaneous zero or non-zero issue, thus making the method robust to the block size.
>
> Although our method still requires a predefined block size, which is necessary because we need to learn a correlation matrix $B_i$ with a determined dimension, the predefined block size acts more like an initial point in DivSBL. Through variance shrinkage, DivSBL can eventually find the true block structure. We have further added visualization experiments related to Figure 4 in Section 5.2 of the paper, where the true signal block sizes are highly non-uniform. You can refer to **Fig. 2 in the global rebuttal PDF**, where regardless of whether we choose a small, medium, or large block size, our method can ultimately identify the true block positions through variance shrinkage, which other block-based methods (like BSBL) do not possess.
>
> ---
>
> ### **Q2 (Sample complexity):**
> This is a very good question. We agree that incorporating too many latent variables could be burdensome for recovery. DivSBL involves $n+g$ latent variables (since the correlation matrix $B_i$ for each block retains only one AR parameter through Toeplitz projection), which is a reasonable number in the Bayesian framework. For instance, the Bayesian Fused Lasso model in [1] involves $2n$ latent variables but is still a classic method in structured sparse learning.
>
> Here, our observation is that BSBL involves $g+1$ latent variables (under the strong constraint where $B_i = B$ for each block). This number of latent variables is too few (even fewer than the $n$ latent variables in SBL), making BSBL unable to effectively capture the prior covariance structure. Since elements within a block share a common variance parameter in the BSBL model, they are estimated to be either all zero or all non-zero simultaneously, making the method highly sensitive to the preset block information, as shown in **Figure 4 of our paper and Fig. 2 of the global rebuttal PDF**.
>
> We have added experiments **in Fig.1 of the global rebuttal PDF** showing the joint impact of observations and noise on the algorithms. Our method remains stable even when observations is close to the number of non-zero elements, supporting our argument: in terms of representing the prior covariance structure, it is not that DivSBL is over-parameterized, but rather that BSBL is under-parameterized.
>
> ---
>
> ### **Q3 (Theoretical results in noisy case):**
> Thanks for your constructive question. For the global minimum, the proof is a generalization of [2], and it seems hard to relax since the equivalent transformations from (35) to (36) in the paper depend on the assumption of noiselessness.
>
> For the local minima, we subsequently adjusted the proof, allowing the method to be extended to noisy scenarios. This also validates our conjecture in the paper. **For a detailed proof, please refer to the global rebuttal**.
>
> ---
>
> ### **Q4 (A closed-form solution in EM):**
> Thanks for your question. As documented in [2], the subproblems of the EM algorithm have close form solutions both in the unconstrained case and under strong constraints. Strong constraints are used to avoid overfitting.
>
> Our contribution lies in proposing a weak constraint framework and provide both explicit and hidden formats for choosing $\psi$. Subproblems of the EM algorithm with explicit constraints require iterative solving via the dual ascent method, while subproblems with hidden constraints have a closed-form solution for $B_i$, as demonstrated in Proposition 3.1. We also provide comparative experiments in Appendix D to demonstrate the effectiveness of algorithm with hidden constraints.
>
> ---
>
> ### **Q5 (PC-SBL):**
> Thank you for your insightful question. The prior in PC-SBL is $p(x_i|\alpha_{i-1},\alpha_{i},\alpha_{i+1}) = \mathcal{N}(x_i;0,(\alpha_{i} + \beta \alpha_{i-1} + \beta \alpha_{i+1})^{-1}).$ This pattern-based method, which uses variance coupling between elements rather than learning correlation, differs from block-based DivSBL. Hence, it is not a special case of DivSBL in Section 2.1.3.
>
> In summary, our proposal of the DivSBL was not aimed at unifying SBL and its variants, nor was it our main contribution. The diversification of variance and correlation matrix had clear motivations and resolved the longstanding sensitivity issue of block-based methods.
>
> ---
>
> ### **Q6 (Error in estimating $B_i$):**
> Thanks for your question. The correlation matrices $B_i$ in the model are latent variables. Given a block sparse signal $x$, it could be generated by arbitrary correlation matrix $B_i$ of any dimension, hence there is no ground truth for $B_i$ in block sparse recovery. The latent variables $B_i$ are introduced here to better exploit block sparse information, but ultimately, only the target signal $x$ is used to measure errors.
>
> ---
>
> References:
>
> [1] Casella G, et al. Penalized regression, standard errors, and Bayesian lassos[J]. 2010.
>
> [2] Zhang Z, Rao B D. Sparse signal recovery with temporally correlated source vectors using sparse Bayesian learning[J]. 2011.

---

### Author Rebuttal · Authors · 2024-08-07

# Global Rebuttal

---

## **1. Experimental Setup in Global Rebuttal PDF**

### **Fig.1:**
The test data in Fig.1 is sourced from the Audioset described in Section 5.3 of the paper. This audio data contains approximately 90 non-zero elements ($K=90$), which constitutes about 20% of the total dimensionality ($N = 480$). Therefore, we start the test measurements with a sampling rate ($M/N$) of a same 20%. In this scenario, $M/K$ is roughly 1 and increases with the sampling rate. Concurrently, the signal-to-noise ratio (SNR) varies gradually from 10 to 50. The phase transition diagram illustrates that DivSBL performs well at more extreme sampling rates and is better suited for lower SNR conditions.

### **Fig.2:**
Figure 2 visualizes the posterior variance learning on the signal from Section 5.2 of the paper to demonstrate DivSBL's ability to adaptively identify the true blocks. The block sizes of the three non-zero blocks are 100, 40, and 30, and the algorithms are tested with preset block sizes of 20 (small), 50 (medium), and 125 (large), respectively, to show how each algorithm learns the blocks when the block structure is misspecified. The first row of each subplot shows the distribution of non-zero elements in the original signal, while the subsequent rows display the posterior variance learned by the comparative algorithms. As expected, DivSBL is able to adaptively find the true block through diversification learning and remains robust to the preset block size, validating our discussion in Figure 2 of the paper.

### **Fig.3 & Fig.4:**
Figures 3 and 4 present experiments on the test signal from Section 5.1 of the paper. Based on the reviewers' valuable suggestions, we have included posterior confidence intervals of the Bayesian methods to better demonstrate the natural advantage of Bayesian models in quantifying uncertainties. Additionally, we have added comparative experiments of recovery errors (NMSE & Correlation) with the horseshoe model, spike & slab LASSO, and hierarchical normal-gamma model for 500 random runs in Fig.4, further illustrating the advantages of DivSBL in block sparse recovery problems.

### **Fig.5:**
The experiment in Fig.5 tests the sensitivity of DivSBL to initialization on the signal data from Section 5.1 of the paper. Initial variances are set to
$\gamma = \eta \cdot \text{ones}(g L, 1)$
and
$\gamma = \eta \cdot \text{rand}(gL, 1)$
with the scale parameter $\eta$ ranging from $1 \times 10^{-1}$ to $1 \times 10^{4}$. The results show that while initialization could affect the convergence speed to some extent, the algorithm's overall convergence is assured.

---

## **2. The Proof of Local Minima in Noisy Scenario**

We sincerely thank the reviewers for your insightful questions, which have driven us to further our theoretical advancements. Your valuable feedback has been instrumental in enhancing the depth and rigor of our work.

We subsequently adjusted the proof, with the core being the direct proof of lines 493-494 in Appendix H of the original paper without relying on Lemma 4.2, allowing the method to be extended to noisy scenarios. We found that the constraint (a.2) with noise can be proved convex with respect to $Z$ and
$\sqrt{\gamma} \otimes \sqrt{\gamma}$  directly.
The proof is as follows:

### **Proof**

We can equivalently transform the constraint  (a.2) with noise into the following:

$$
Z \succeq \Phi \Sigma_0 \Phi^T + \beta^{-1} I
\Longleftrightarrow \forall \omega \in \mathbb{R}^m, \quad \omega^T Z \omega \geq \omega^T \Phi \Sigma_0 \Phi^T \omega + \beta^{-1} \omega^T \omega.
$$

The LHS $\omega^T Z \omega$ is linear with respect to $Z$. And for the RHS, $\omega^T \Phi \Sigma_0 \Phi^T \omega + \beta^{-1} \omega^T \omega = q^T \Sigma_0 q + \beta^{-1} \omega^T \omega$, where $q = \Phi^T \omega$, and $\Sigma_0$ can be reformulated as:

$$
\Sigma_0 = \text{diag}(\sqrt{\gamma_{11}}, \ldots, \sqrt{\gamma_{gL}}) \tilde{B} \text{diag}(\sqrt{\gamma_{11}}, \ldots, \sqrt{\gamma_{gL}})
= \begin{bmatrix}
{\gamma_{11}} \tilde{B}\_{11} & \sqrt{\gamma_{11}} \sqrt{\gamma_{12}} \tilde{B}\_{12} & \ldots & \sqrt{\gamma_{11}} \sqrt{\gamma_{gL}} \tilde{B}\_{1N} \\\\
\vdots & \vdots & & \vdots \\\\
\sqrt{\gamma_{gL}} \sqrt{\gamma_{11}} \tilde{B}\_{N1} & \sqrt{\gamma_{gL}} \sqrt{\gamma_{12}} \tilde{B}\_{N2} & \ldots & {\gamma_{gL}} \tilde{B}\_{NN}
\end{bmatrix}
$$

Therefore, RHS:

$$
\sum_{i=1}^{N} \sum_{j=1}^{N} q_i q_j \tilde{B}\_{ij} \sqrt{\gamma_{i}} \sqrt{\gamma_{j}} + \beta^{-1} \omega^T \omega = \text{vec}(q q^T \odot \tilde{B})^T (\sqrt{\gamma} \otimes \sqrt{\gamma}) + \beta^{-1} \omega^T \omega
$$

which is linear with respect to $\sqrt{\gamma} \otimes \sqrt{\gamma}$.

In conclusion, $Z \succeq \Phi \Sigma_0 \Phi^T + \beta^{-1} I$ is convex with respect to $Z$ and $\sqrt{\gamma} \otimes \sqrt{\gamma}$, which can conclude the proof in Theorem 4.4 in the noisy case ($\forall \beta$).

---

### Decision · Program_Chairs · 2024-09-25

**Decision:**

Accept (poster)

**Comment:**

(*Note: the meta-review has been updated and finalized.*)

Thanks for your submission to NeurIPS! This is a borderline case, where
the reviewers failed to reach a consensus. Considering the overall
comments, feedback and further discussions with the SAC, the
recommendation from the AC is accept as poster.

The AC inclines to support the acceptance of this paper due to the
following points:

- A sound generalization of priors for SBL is proposed, and its
  inference algorithm is studied in theory. This generalization indeed
  broadens the applicability range of SBL, which may have profound
  potentials in practice.

- The weakness points on the fixed and known block size and position in
  review comments could be misunderstanding, probably due to the
  presentation that should be further improved. The experiments have
  considered the case where both are unknown.

- The criticism on no guarantee on exact inference of sparsity in theory
  could be the common issue in SBL or even sparse learning in general.
  Except the basic case like LASSO, such strict theoretical guarantees
  are mostly particularly challenging, which are admittedly desired but
  may take longer to be developed.

- Most technical questions have been answered in the rebuttal, although
  some reviewers did not follow up further discussions.

However, the authors need to take into account these negative feedback
and comments carefully to improve the paper:

- Figures are not satisfactory. The sizes of figures and fonts and their
  positions in paper should be carefully designed: Fig.3 and 5 are
  barely readable due to their particular tiny fontsize; the right
  subfigure in Fig.4 uses the much larger fontsize than the others. The
  authors need to take it serious in preparing the camera-ready. The
  current quality fails to meet the average of NeurIPS.

- We strongly suggest the authors to carefully organise/tidy the math
  symbols, and clearly tell what variables are fixed/variable,
  known/unknown. Apparently some reviewers were confused with these
  different setups of hyperparameters, etc. Indeed there are
  complications in the notation system due to the method itself, hence
  your careful writing is particularly favoured to ease understanding
  for readers.

- The theoretical results in Sec.4 may be better motivated and
  explained. A list of theories plainly addressed in math may not be
  much useful for readers to better understand and learn the boundary
  and strengths/weaknesses of your method.

We thank the authors for your further efforts on preparing the camera-ready and  your contributions to NeurIPS!

Bests,

The AC